# Learning Substructure Invariance for Out-of-Distribution Molecular Representations

**Nianzu Yang, Kaipeng Zeng, Qitian Wu, Xiaosong Jia, Junchi Yan**[*]
Department of Computer Science and Engineering
MoE Key Lab of Artificial Intelligence
Shanghai Jiao Tong University
`{yangnianzu,zengkaipeng,echo740,jiaxiaosong,yanjunchi}@sjtu.edu.cn`

## Abstract

Molecule representation learning (MRL) has been extensively studied and current methods have shown promising power for various tasks, e.g., molecular property prediction and target identification. However, a common hypothesis of existing methods is that either the model development or experimental evaluation is mostly based on i.i.d. data across training and testing. Such a hypothesis can be violated in real-world applications where testing molecules could come from new environments, bringing about serious performance degradation or unexpected prediction. We propose a new representation learning framework entitled MoleOOD to enhance the robustness of MRL models against such distribution shifts, motivated by an observation that the (bio)chemical properties of molecules are usually invariantly associated with certain privileged molecular substructures across different environments (e.g., scaffolds, sizes, etc.). Specifically, We introduce an environment inference model to identify the latent factors that impact data generation from different distributions in a fully data-driven manner. We also propose a new learning objective to guide the molecule encoder to leverage environment-invariant substructures that more stably relate with the labels across environments. Extensive experiments on ten real-world datasets demonstrate that our model has a stronger generalization ability than existing methods under various out-of-distribution (OOD) settings, despite the absence of manual specifications of environments. Particularly, our method achieves up to 5.9% and 3.9% improvement over the strongest baselines on OGB and DrugOOD benchmarks in terms of ROC-AUC, respectively. Our source code is publicly available at `https://github.com/yangnianzu0515/MoleOOD`.

## 1 Introduction

Predicting molecular properties plays an important role in many related applications like drug discovery [13] and material design [51]. These professional tasks conventionally take great efforts by experts e.g. in chemistry and pharmacology. Recent years have witnessed inspiring breakthroughs on building effective machine learning models for scientific discovery, and solid progress has been made along the avenue of ML-based molecule representation learning (MRL). In general, MRL aims at embedding a molecule into a vector in latent space as a foundation model, on top of which the learned representations could be used for a variety of downstream tasks, such as target identification [69], retrosynthetic analysis [65], search of antibiotics [54], virtual screening [40] for drug discovery, etc.

The challenge, however, is that existing MRL methods are mostly based on an underlying hypothesis that training and testing molecules are independently sampled from an identical environment, yet

---

[*]Junchi Yan is the correspondence author who is also with Shanghai AI Laboratory.

36th Conference on Neural Information Processing Systems (NeurIPS 2022).

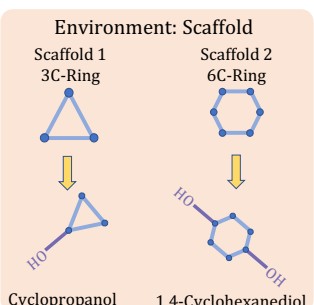 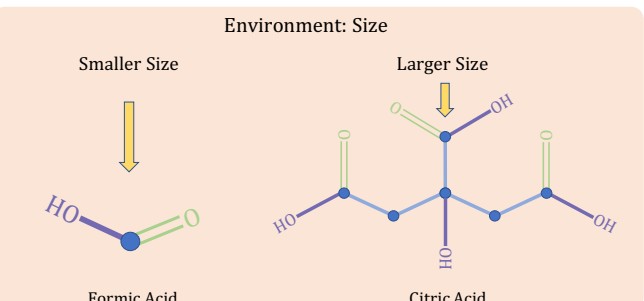

Figure 1: Two examples. **Left:** the shared substructure hydroxy $(-OH)$ invariantly contributes to the water solubility of the two molecules which contain different scaffolds, i.e. sampled from different environments by definition. **Right:** the water solubility of the two molecules with different sizes can be attributed to the shared substructure carboxy $(-COOH)$ invariantly, where different sizes are regarded as indicators to define different environments.

real-world environments are often dynamic and uncertain, which requires the model to effectively handle distribution shifts. In fact, the available experimental molecule data are rather limited while the candidate molecules to be tested are often diverse, coming from unknown environments. Taking the virtual screening [40] as an example (which is a common protocol in drug discovery and usually for target identification), the prediction model is typically trained on some known target proteins. However, some unpredictable events like COVID-19 may occur, bringing new targets from unknown distributions. Similar scenarios where training and testing data are sampled from different distributions are common in real world, posing an urgency for strengthening current MRL methods regarding out-of-distribution (OOD) generalization [41, 7, 44].

Existing methods devised for out-of-distribution generalization mostly focus on Euclidean data such as images, while few endeavors OOD generalization on non-Euclidean data [61, 39]. In particular, molecules, as a kind of typical non-Euclidean data, i.e., graph-structured data, is different from visual data in nature. The work [27] point out that existing OOD models [3, 55, 18, 70, 50] fail to exhibit significant improvement on MRL tasks against distribution shifts and even the simple Empirical Risk Minimization (ERM) [56] method outperforms these latest methods, which is also empirically verified by [16]. We aim to develop an OOD method tailored for molecules to solve the OOD generalization problem on MRL in this paper.

We incorporate an effective prior in the molecule domain into our model design: the (bio)chemical properties of a molecule are usually associated with a few privileged molecular substructures, which has been consistently shown by studies [31, 48, 72, 29] across bio-informatics, pharmacy, and data mining. The common practice specifies environments as some prominent information of the molecules e.g. scaffold pattern [32, 23] and molecule size [27]. Fig. 1 provides two illustrative examples. Let's first take a look at the left example, where two molecules *Cyclopropanol* $(C_3H_6O)$ and *1,4-Cyclohexanediol* $(C_6H_{12}O_2)$ contain different scaffold patterns[2]: the former is *3C-ring* and the latter is *6C-ring*. Thus, the data-generating environments and the induced distributions which these two molecules are sampled from can be considered different [23]. Though sampled from different distributions, they are both readily soluble in water due to the invariant substructure hydroxy [24] shared across different environments. As for the example on the right of Fig. 1, the sizes of two molecules *Formic Acid* $(CH_2O_2)$ and *Citric Acid* $(C_8H_8O_7)$ differ a lot. Consequently, they can also be considered as being sampled from different environments. Owing to the shared invarint substructure carboxy $(-COOH)$, they are both readily soluable in water, too. Hence, a promising paradigm would be to learn the causal data-generating invariance from the substructures across environments, regarding a certain property, for the OOD generalization purpose.

Another important observation for consideration is that existing specifications for environments are often handcrafted or rule-based and not structured, which could provide insufficient information for capturing the fundamental relations across domains from the casual data-generating perspective.

---

[2]As a 2-D structural molecular framework [5], the scaffold reduces the chemical structure of a molecule to its core components, which can be obtained by removing side chains and only reserving the rings and parts connecting rings [67]. The scaffold can be an indicator to define a specific environment [32, 23].

Besides, some studies [27, 16] show that directly utilizing such environment labels as input when adapting existing OOD generalization methods to MRL tasks can be problematic. Furthermore, manual specifications of environments may be unavailable in reality. Hence, we aim to develop a label-free model that does not rely on the above ad-hoc environment labels. As shown later, our model can infer the environment labels in an unsupervised manner, namely for environment clustering.

To achieve robust molecule representation for OOD generalization and overcome potentially unreliable environment labels, we devise a new MRL framework without explicitly using the environment label information. We first formulate OOD generalization for molecular property prediction by introducing a latent variable for environments that affect the data generation. Then we analyze the essential cause behind the failure of existing MRL models and propose a new learning scheme based on the invariance principle [47, 44, 3, 61]. The training procedures contain two steps: 1) optimize an environment inference model from training data; 2) optimize a molecule encoder and a predictor. Our general framework can integrate existing GNN backbones and achieve improvements on four OGB molecular property prediction tasks [23], as shown in our experimental results. As for a newly released benchmark for drug-oriented OOD learning [27], even without access to environment labels, our method can still outperform state-of-the-art models that rely on environment labels for training in five out of six datasets. **The contributions of this paper are:**

- We formulate the out-of-distribution (OOD) generalization problem for molecule representation learning (MRL), by particularly incorporating an important observation that the substructure of molecule can convey invariant casual information across environments, regarding certain property prediction tasks. To our best knowledge, this is the first work that formulates the OOD problem in MRL background and proposes to leverage the invariance principle which opens a new perspective for handling substructure-aware distribution shifts.

- Under the environment-invariance principle with specific substructure invariance priors, we propose a new learning objective to learn robust representations. In particular, our model does not require environment labels which in fact can be noisy and unreliable, but instead achieve environment inference in an unsupervised manner. This design endows our model with practical applicability for molecular OOD learning where the manual specifications of the environments are often unavailable.

- We conduct extensive experiments on ten public datasets. Results demonstrate that our model yields consistent and significant improvements over various existing MRL methods as backbones and also achieves competitive or even superior prediction compared to state-of-the-art models tailored to OOD learning with environment labels used as extra inputs in both training and testing. Particularly, our method achieves up to 5.9% higher ROC-AUC on public OGB molecular property prediction benchmarks than the counterpart model trained with traditional objective. Besides, for drug-oriented benchmarks DrugOOD, when environment labels are not used, our model still outperforms several SOTA approaches tailored for general OOD learning (using environment labels as extra training information) by up to 3.9% w.r.t. ROC-AUC.

## 2 Backgrounds and Related Works

**Out-of-Distribution Generalization.** Deep neural networks are prone to suffering significant performance degradation under distribution shifts, motivating a surge of works on OOD generalization. Recent studies [49, 3, 9, 61] assume that there is a potential environment variable $\mathbf{e}$ accounting for the distribution shift between the training and testing data. In general cases the goal is to predict the target label $\mathbf{y}$ given the associated input $\mathbf{x}$. Then, the OOD problem could be formally formulated as:

$$\min_f \max_{e \in \mathcal{E}} \mathbb{E}_{(x,y) \sim p(\mathbf{x},\mathbf{y}|\mathbf{e}=e)}[l(f(x),y)|e],\tag{1}$$

where $\mathcal{E}$ denotes the support of environments, $f(\cdot)$ is the prediction model and $l(\cdot,\cdot)$ represents a loss function. Notice that $\mathbb{E}_{(x,y) \sim p(\mathbf{x},\mathbf{y}|\mathbf{e}=e)}[l(f(x),y)|e]$ is called the **risk function** under a given environment $e$ and denoted as $\mathcal{R}_e(\mathbf{x}^e, \mathbf{y}^e)$ [33].

**Invariant Learning.** There is an emerging line of research [49, 3, 10, 12] regarding invariant predictor learning, for solving the OOD generalization problem. These methods propose to find an invariant predictor that could uncover invariant relationships between inputs and targets across all environments [33]. The invariant predictor aims to learn an invariant representation satisfying such a **invariance principle**: 1) **sufficiency**: shows sufficient predictive power for the target, 2) **invariance**:

contributes to equal (optimal) performance for the downstream tasks across all environments. Recent works adopt the invariance principle as a cornerstone for handling various distribution shifts on semi-structured data like graphs [61, 6] and sequences [66]. To our knowledge, our work is a pioneering attempt that leverages the invariance principle and incorporate useful domain knowledge for handling molecular graph classification tasks under distribution shifts.

**Molecule Representation Learning.** Existing molecule representation learning methods can be classified into two categories. The first is SMILES-based methods where SMILES refers to Simplified Molecular Input Line Entry System [2]. They use language models to process the textual representation (SMILES) of a molecule, for example, Transformer [57] or BERT [15]. SMILES is a linear encoding for molecules and highly depends on the traverse order of molecule graphs. Therefore its expressiveness is limited for problems like medication recommendation which we believe calls for fine-grained molecular structure extraction. Beyond the above linear encoding protocol, structure-based methods are also developed, which can be further classified into fingerprint-based and graph neural networks (GNN)-based methods. The molecular fingerprint techniques date back to the Morgan fingerprints [43]. However, those fingerprint-based methods are often handcrafted and not trained in an end-to-end fashion [26]. Since molecules can be viewed as structured graphs, methods tailored for graph data, such as graph neural networks [28, 19, 24] or graph transformers [17, 62] can be used to learn molecule representation.

Existing general OOD methods [3, 55, 18, 70, 50] are not tailored to such non-Euclidean structured data, i.e. molecules. In several recent works on molecule property classification tasks [71, 59], the importance of molecular substructures has been emphasized and such inductive bias is incorporated into the design of those models. However, they are still based on the i.i.d. assumption and do not leverage those invariant substructure across different environments to achieve robust representations. In this paper, we propose a general framework orthogonal to these MRL studies to bridge OOD and MRL, which can adopt any existing MRL methods as the backbone to improve their robustness.

## 3 Methodology

### 3.1 Problem Formulation

We propose a OOD generalization framework tailored for molecule representation learning, entitled MoleOOD. All the random variables and the corresponding realizations are denoted as bold and thin letters, respectively. We first formulate the OOD generalization problem for MRL.

**OOD Generalization Problem on Molecule Representation Learning.** A molecular graph can be represented as $G = (V, E)$, where $V$ is the graph's node set corresponding to atoms constituting the molecule and $E$ denotes the graph's edge sets corresponding to chemical bonds. The training and testing molecule graph datasets are denoted as $\mathcal{G}^{train} = \{(G_i, y_i)\}_{i=1}^{N^{train}}$ and $\mathcal{G}^{test} = \{(G_i, y_i)\}_{i=1}^{N^{test}}$. Notice that the test dataset is drawn outside the distribution of the training dataset. The goal of molecule representation learning task is to predict the target label $\mathbf{y}$ given the associated input molecule $\mathbf{G}$. Based on Eq. 1, we can formulate the OOD problem on MRL tasks as:

$$\min_{f} \max_{e \in \mathcal{E}} \mathbb{E}_{(G_i, y_i) \sim p(\mathbf{G}, \mathbf{y}|\mathbf{e}=e)}[l(f(G_i), y_i)|e].\tag{2}$$

The difficulty of this problem is that the training data only cover very limited environments in $\mathcal{E}$ while the model is expected to perform well on all the environments.

We elaborate our approach in the context of molecule property classification tasks in this paper. Existing MRL methods do not differentiate invariant and spurious substructures. Hence, the spurious correlations between irrelevant substructures and the target label will be encoded to learned molecular representations. When tested on unseen environments, the downstream classifier will be easily misled by these spurious correlations [3]. With the knowledge that (bio)chemical properties of a molecule are usually associated with a few privileged substructures [31, 48, 72, 29], we aim to suppress such spurious correlations and leverage environment-invariant substructures that more stably relate with the labels across environments to learn invariant molecular representations. Notice that the learned invariant molecular representations should satisfy the invariance principle mentioned in Sec. 2. We next introduce our method formally and then give the instantiation of our model.

## 3.2 Model Formulation

The framework contains two parts, the fronted molecule encoder $\Phi$ for learning an "invariant representation" of the input molecule graph and the back-end predictor $\omega$ for final prediction. Solving the formulation in Eq. 2 directly is intractable in practice since we cannot know all the environments, i.e, obtain a complete support set $\mathcal{E}$. We resort to minimizing the expectation of risks from different environments known in the training data,

$$\min_{\omega, \Phi} \mathbb{E}_{\mathbf{e}}[\mathcal{R}_{\mathbf{e}}(\mathbf{G}^{\mathbf{e}}, \mathbf{y}^{\mathbf{e}})], \text{ s.t. } \mathbf{y} \perp\!\!\!\perp \mathbf{e} \mid \Phi(\mathbf{G}), \tag{3}$$

where $f = \omega \circ \Phi$ and $\perp\!\!\!\perp$ denotes probabilistic independence. All learnable parameters of the molecule encoder $\Phi$ and the predictor $\omega$ are included in $\theta$. Different from Eq. 2, we add an extra invariance constraint $\mathbf{y} \perp\!\!\!\perp \mathbf{e} \mid \Phi(\mathbf{G})$, which is used to suppress spurious correlations [10]. Since assessing causality is challenging, we could rethink the problem on the basis of information theory. Recall that we hope to let the molecule encoder leverage environment-invariant substructures and learn a molecular representation $\Phi(G)$ given a molecule $G$. Our goal is to maximize the predictive power of $\Phi(\mathbf{G})$ on $\mathbf{y}$, which can be measured by mutual information between $\Phi(\mathbf{G})$ and $\mathbf{y}$. Meanwhile, probabilistic independence between $\mathbf{y}$ and $\mathbf{e}$ given $\Phi(\mathbf{G})$ can be achieved via minimizing their mutual information. For convenience, we denote $\Phi(\mathbf{G})$ as $\mathbf{z}$ and Eq. 3 can be approximately solved by:

$$\max_{\omega, \Phi} I(\mathbf{z}; \mathbf{y}), \text{ s.t. } \min_{\omega, \Phi} I(\mathbf{y}; \mathbf{e}|\mathbf{z}). \tag{4}$$

Treating the outputs of $\omega$ and $\Phi$ as distribution $q_{\theta}(\mathbf{z}|\mathbf{G})$ and $q_{\theta}(\mathbf{y}|\mathbf{z})$ respectively, Eq. 4 can be specified as:

$$\max_{q_{\theta}(\mathbf{y}|\mathbf{z}), q_{\theta}(\mathbf{z}|\mathbf{G})} I(\mathbf{z}; \mathbf{y}), \text{ s.t. } \min_{q_{\theta}(\mathbf{y}|\mathbf{z}), q_{\theta}(\mathbf{z}|\mathbf{G})} I(\mathbf{y}; \mathbf{e}|\mathbf{z}). \tag{5}$$

Now, we have arrived at a clearer but still intractable optimization objective. Before specifying the practical instantiation of Eq. 5, let's discuss on the environment variable $\mathbf{e}$ first.

In practice, due to the non-trivial efforts to label the molecular environments, manual specifications of the environments may be unavailable in many cases. We may directly label molecules to different environments in terms of their scaffolds when the environment label is unavailable. But this is unreasonable in practice, because the final total environment number will be too large. Taking the dataset HIV for molecule property prediction tasks released by Open Graph Benchmark [23] as an example, OGB uses scaffold to split the molecules into different environments. Assuming that we regard each scaffold as an environment directly, $41,127$ molecules in HIV are partitioned into $19,076$ environments (see details in Appendix D). This environment count is much larger than other OOD datasets from other domains, e.g. Camelyon17[3] [4], CivilComments[4] [8], etc. Even though some datasets may provide manual specifications of environments, the environment counts are also too large, which is unfriendly to existing OOD models [27, 16]. Therefore, we propose to design an environment-inference model $\psi$ to partition the molecule into different environments with a relatively smaller environment count. We denote the environment count as a hyper-parameter $k$.

Given prior $p(\mathbf{e}|\mathbf{G})$, we need to maximize the log likelihood of $p_{\tau}(\mathbf{y}|\mathbf{G})$ and then obtain the posterior $p_{\tau}(\mathbf{e}|\mathbf{G}, \mathbf{y})$, which are parameterized by $\tau$. Since there is no analytical solutions to the true posterior, here we use variational inference (VI) to approximate it. Specifically, we introduce a variational distribution $q_{\kappa}(\mathbf{e}|\mathbf{G}, \mathbf{y})$ parameterized by $\kappa$ to approximate $p_{\tau}(\mathbf{e}|\mathbf{G}, \mathbf{y})$.

**Proposition 1.** *The Evidence Lower BOund (ELBO) of the observed molecule graph and corresponding label tuple* $(G, y)$: $\mathcal{L}(\tau, \kappa; (G, y)) = \mathbb{E}_{q_{\kappa}}[\log p_{\tau}(y|G, e)] - D_{KL}(q_{\kappa}(e|G, y) \| p_{\tau}(e|G))$.

Our goal is to minimize the Kullback-Leibler (KL) divergence between $q_{\kappa}(\mathbf{e}|\mathbf{G}, \mathbf{y})$ and $p_{\tau}(\mathbf{e}|\mathbf{G}, \mathbf{y})$, i.e. $D_{KL}(q_{\kappa}(\mathbf{e}|\mathbf{G}, \mathbf{y}) \| p_{\tau}(\mathbf{e}|\mathbf{G}, \mathbf{y}))$, which is equivalent to maximizing the ELBO in Proposition 1. Then, the objective used to train this environment-inference model is transformed to:

$$\mathcal{L}_{elbo}(\tau, \kappa; \mathcal{G}) = \frac{1}{|\mathcal{G}|} \sum_{(G,y) \in \mathcal{G}} [\mathbb{E}_{q_{\kappa}}[\log p_{\tau}(y|G, e)] - D_{KL}(q_{\kappa}(e|G, y) \| p(e|G))]. \tag{6}$$

---

[3]Camelyon17 is for tumor prediction, partitioning $455,954$ issue slides into $5$ environments.

[4]CivilComments is for toxicity prediciton, partitioning $448,000$ online comments into $16$ environments.

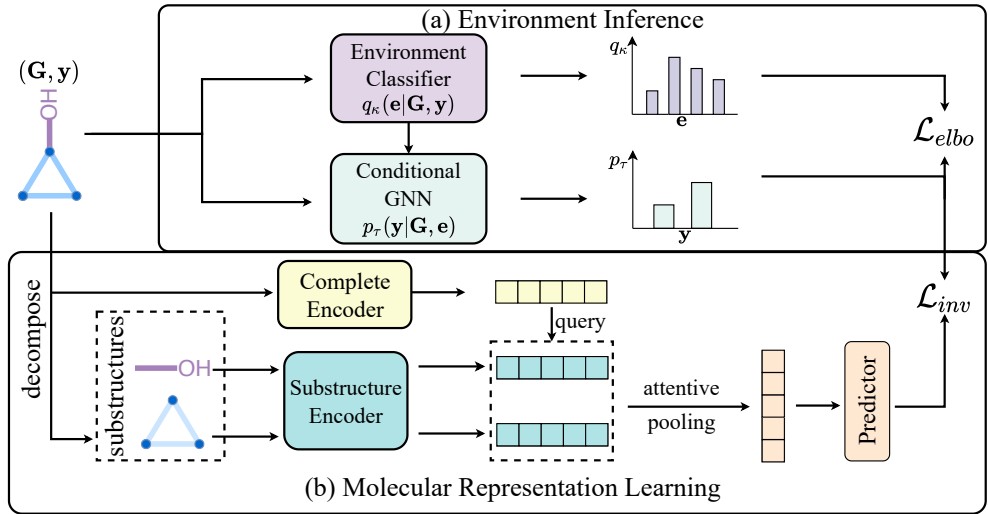

Figure 2: Overview of our model. The whole training procedure is divided into two stages: 1)Optimize the environment-inference model. Given an input molecule $(\mathbf{G}, \mathbf{y})$, we first infer the latent environment variable $\mathbf{e}$. This stage is trained under the guidance of $\mathcal{L}_{elbo}$. 2) Optimize the molecule encoder and the final predictor guided by $\mathcal{L}_{inv}$.

Let's look back to the objective given in Eq. 5 and give an equivalent tractable objective in practical instantiation, which involves the environment-inference model defined above:

$$\mathcal{L}_{inv}(\theta; \mathcal{G}, \tau) = \underbrace{\frac{1}{|\mathcal{G}|} \sum_{(G,y) \in \mathcal{G}} \left| \log q_\theta(y|G) - \mathbb{E}_{p(\mathbf{e}|\mathbf{G})}[\log p_\tau(y|G, e)] \right|}_{①} + \underbrace{\beta \mathbb{E}_{\mathbf{e}} \left[ \frac{1}{|\mathcal{G}^e|} \sum_{(G,y) \in \mathcal{G}^e} [-\log q_\theta(y|G)] \right]}_{②}, \quad (7)$$

where $\mathcal{G}^e$ consists of the pairs of molecular graph $G$ and corresponding label $y$ under environment $e$.

**Theorem 1.** *With $q_\theta(\mathbf{y}|\mathbf{z})$ treated as a variational distribution, minimizing term* ① *in Eq. 7 contributes to* $\min_{q_\theta(\mathbf{y}|\mathbf{z}), q_\theta(\mathbf{z}|\mathbf{G})} \mathrm{I}(\mathbf{y}; \mathbf{e}|\mathbf{z})$, *letting $\mathbf{z}$ show equal performance for the downstream tasks across all environments, i.e. $p(\mathbf{y}|\mathbf{z}, \mathbf{e}) = p(\mathbf{y}|\mathbf{z})$.*

**Theorem 2.** *Regarding $q_\theta(\mathbf{y}|\mathbf{z})$ as a variational distribution, minimizing term* ② *in Eq. 7 equals to* $\max_{q_\theta(\mathbf{y}|\mathbf{z}), q_\theta(\mathbf{z}|\mathbf{G})} \mathrm{I}(\mathbf{z}; \mathbf{y})$, *letting $\mathbf{z}$ show sufficient predictive power for downstream tasks.*

Serving as theoretical justifications, Th. 1 and Th. 2 reveal that optimizing the objective in Eq. 7 forces the learned representation $\mathbf{z}$ to satisfy the invariance principle mentioned in Sec. 2, thus ensuring a valid solution for OOD problem defined in Eq. 2. Due to the limited space, the detailed proofs can be found in Appendix B.

### 3.3 Model Instantiations and Training

**Environment-inference Module.** For the approximate posterior model $q_\kappa(\mathbf{e}|\mathbf{G}, \mathbf{y})$, in principle we should design a module, entitled **Environment Classifier**, that takes $(\mathbf{G}, \mathbf{y})$ as the input and outputs the probabilistic distribution of $\mathbf{e}$. We use a Graph Isomorphism Network (GIN) [63], to learn a graph representation given $\mathbf{G}$. Then, the concatenation of this graph representation and label vector is fed to a feed-forward network to obtain a probabilistic distribution with regard to $\mathbf{e}$. We could set the prior $p_\tau(\mathbf{e}|\mathbf{G})$ to *Uniform distribution* or *Gaussian distribution*. As for $p_\tau(\mathbf{y}|\mathbf{G}, \mathbf{e})$, we also choose a GNN model (e.g. GIN) followed by a softmax activation function to model it. We call this module **Conditional GNN** because it conditions on $\mathbf{e}$. It takes $(\mathbf{G}, \mathbf{e})$ as the input and outputs the probabilistic distribution of $\mathbf{y}$.

**The Molecule Encoder & The Final Predictor.** Recall that we aim to learn an invariant substructure-aware molecular representation. Given a molecule $\mathbf{G}$, we can choose any molecule representation learning method to learn a representation $\mathbf{r_G}$ for the complete molecular graph. This part is entitled **Complete Encoder**. Meanwhile, we decompose the input molecule into a set of chemical

substructures using a molecule segmentation method, e.g. *breaking retrosynthetically interesting chemical substructures* (BRICS) [14], which is available as an API in RDKit [34]. For each substructure, we consider using a simple GNN to learn a corresponding representation. We call this GNN **Substructure Encoder**. Then, considering $\mathbf{r_G}$ as a query with regard to substructures, we operate attentive pooling on these substructure representations to obtain a new substructure-aware molecular representation. We then use this substructure-aware representation for downstream task. Guided by our proposed learning objective, we can encode some invariant relationships between certain substructures and target properties into this representation. The Complete Encoder, the Substructure encoder and the attentive pooling operation constitute our **Molecule Encoder** $\Phi$. As for the **Predictor** $\omega$, we implement it with a multi-layer perceptron, followed by a softmax function. The overview of our model is demonstrated in Fig. 2.

**Training.** We adopt a simple yet efficient two-stage training strategy to search for optimal parameters and the training procedure of our method is summarized in Algorithm 1:

1) **optimizing the environment-inference model:** $\kappa^*, \tau^* \leftarrow \arg\max_{\kappa,\tau} \mathcal{L}_{elbo}(\tau, \kappa; \mathcal{G}^{train})$.

2) **optimizing the molecule encoder and the predictor:** $\theta^* \leftarrow \arg\min_\theta \mathcal{L}_{inv}(\theta; \mathcal{G}^{train}, \tau)$.

---

**Algorithm 1:** The training procedure.

**Input:** Dataset $\mathcal{G}^{train} = \{(G_i, y_i)\}_{i=1}^{N^{train}}$; Number of training epochs for environment inference module $E_1$; Number of training epochs for the molecule encoder and the predictor $E_2$; Batch size $B$.
**Output:** Trained parameters $\theta$.

1 Initialize parameters $\theta, \tau$ and $\kappa$;
2 **for** $i \leftarrow 1$ **to** $E_1$ **do**
3      Sample data batches $\mathcal{B} = \{\mathcal{G}_1, \mathcal{G}_2, \dots, \mathcal{G}_k\}$ from $\mathcal{G}^{train}$ with batch size $B$;
4      **for** $j \leftarrow 1$ **to** $k$ **do**
5          Compute batch loss $\mathcal{L}_{elbo}(\tau, \kappa; \mathcal{G}_j)$ according to Eq. 6;
6          Backpropagate $-\mathcal{L}_{elbo}$ and optimize parameters $\tau, \kappa$;

7 Freeze the parameters $\kappa, \tau$;
8 **for** $i \leftarrow 1$ **to** $E_2$ **do**
9      Sample data batches $\mathcal{B} = \{\mathcal{G}_1, \mathcal{G}_2, \dots, \mathcal{G}_k\}$ from $\mathcal{G}^{train}$ with batch size $B$;
10      **for** $j \leftarrow 1$ **to** $k$ **do**
11          Determine the environment of each sample $(G, y)$ in $\mathcal{G}_k$ by $\arg\max_e q_\kappa(e|G, y)$;
12          Compute batch loss $\mathcal{L}_{inv}(\theta; \mathcal{G}_k, \tau)$ according to Eq. 7;
13          Backpropagate $\mathcal{L}_{inv}$ and optimize parameters $\theta$;

14 Output the parameters $\theta$;

---

## 4 Experiments

Experiments are performed on 10 benchmark datasets and repeated 5 times with mean and standard deviation reported, running on a machine with i9-10920X CPU, RTX 3090 GPU and 128G RAM.

### 4.1 Datasets and Setups

**Datasets and protocols.** The four datasets **BACE**, **BBBP**, **SIDER** and **HIV**, are from by Open Graph Benchmark (OGB) [23]. We use the default train/val/test split with ratio 8:1:1. Each split contains a set of scaffolds (almost) different to each other. Hence we believe that to a certain degree, it provides an OOD test-bed as different scaffold often suggest different data-generation environments. The other six datasets are generated by the dataset curator provided by DrugOOD [27]. DrugOOD provides more diverse splitting indicators than OGB, including assay, scaffold and size. To comprehensively evaluate the performance of our method under different environment definitions, we adopt these three different splitting schemes on categories IC50 and EC50 provided in DrugOOD. Then we obtain six datasets, **EC50-∗** and **IC50-∗**, where the suffix ∗ specifies the splitting scheme i.e. **IC50/EC50-assay/scaffold/size**. Notice that only the six datasets from DrugOOD provide manual specified environment labels. Refer to Appendix D for more details of datasets.

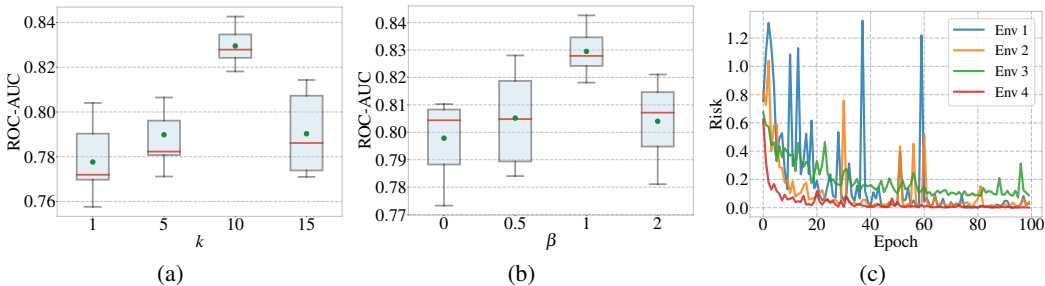

Figure 3: (a) Varying the specified environment number $k$. (b) Varying the trading-off parameter $\beta$ in Eq. 7. (c) Risk curves of environments in the training process. All results are from 'GraphSAGE + ours.' on BACE dataset.

Table 1: Performance comparison with baselines on 4 out-of-distribution molecular property prediction datasets from Open Graph Benchmark (OGB) [23] in terms of ROC-AUC (%), namely, BACE, BBBP, SIDER and HIV. The best and the runner-up results are highlighted in **bolded** and underlined respectively. We emphasize the comparison against '∗ **+ virtual node**', a variant of the original method augmented by an additional node connecting to all nodes in the raw graphs [19, 25, 38].

| Methods | BACE | BBBP | SIDER | HIV |
|---|---|---|---|---|
| **GCN** [30] | $80.01 \pm 3.49$ | $67.92 \pm 1.07$ | $58.90 \pm 1.30$ | $76.35 \pm 2.01$ |
| **GCN + virtual node** | $\overline{77.51 \pm 3.07}$ | $\underline{68.19 \pm 1.86}$ | $\underline{60.71 \pm 1.34}$ | $\overline{75.76 \pm 2.21}$ |
| **GCN + ours.** | $\mathbf{84.33 \pm 1.07}$ | $\mathbf{70.62 \pm 0.99}$ | $\mathbf{63.38 \pm 0.67}$ | $\mathbf{77.73 \pm 0.76}$ |
| **GIN** [63] | $77.83 \pm 3.15$ | $\underline{66.93 \pm 2.31}$ | $59.05 \pm 1.47$ | $76.58 \pm 1.02$ |
| **GIN + virtual node** | $\underline{79.64 \pm 2.02}$ | $66.77 \pm 0.95$ | $\underline{59.12 \pm 0.95}$ | $\underline{77.11 \pm 0.96}$ |
| **GIN + ours.** | $\mathbf{81.09 \pm 2.03}$ | $\mathbf{69.84 \pm 1.84}$ | $\mathbf{61.63 \pm 1.08}$ | $\mathbf{78.31 \pm 0.24}$ |
| **GraphSAGE** [21] | $77.41 \pm 1.19$ | $\underline{70.58 \pm 0.58}$ | $58.00 \pm 0.95$ | $76.98 \pm 1.13$ |
| **GraphSAGE + virtual node** | $\underline{78.34 \pm 2.08}$ | $69.29 \pm 0.99$ | $\underline{59.48 \pm 1.37}$ | $\underline{77.28 \pm 1.53}$ |
| **GraphSAGE + ours.** | $\mathbf{82.95 \pm 0.85}$ | $\mathbf{71.02 \pm 0.75}$ | $\mathbf{61.09 \pm 0.28}$ | $\mathbf{79.39 \pm 0.51}$ |

**Metric.** As the concerned property prediction tasks all relate to classification, we report the ROC-AUC score which is also in line with previous MRL works [68, 64, 58].

**Baselines.** Ideally, any MRL method can be adapted into our method as backbone to improve their generalization ability against distribution shifts. We adapt three backbones: **GCN** [30], **GIN** [63] and **GraphSAGE** [21] into our method. We compare the adapted version with the original method. We also compare against another augmented version "**+ virtual node**" [19, 25, 38]. Furthermore, we compare our method with six OOD generalization methods on MRL tasks: **ERM** [56], **IRM** [3], **DeepCoral** [55], **DANN** [18], **MixUp** [70] and **GroupDro** [50]. Due to the fact that most of these methods require the manual specification of environments in dataset, we report this comparison on datasets from DrugOOD only. Each of the method is configured using the same parameters reported in the original paper or selected by grid search. For the sake of fairness, the embedding size of all methods are set to be equal in comparison. We specify the training details in the Appendix C.

## 4.2 Performance Comparison

**Improvements to existing MRL methods.** As demonstrated in Table 1, baselines obtain consistent improvements after adapted to our methods across all the four datasets released by OGB in terms of ROC-AUC. Our method also beats the augmented version, "+ virtual node" , of baselines on all datasets, i.e. adding a virtual node. The results indicate that, orthogonal to prior studies on MRL, our method is a general framework which can incorporate existing MRL methods and improve their generalization ability for OOD data. We attribute the superior performance of our method in molecular properties predictions under OOD setting to that, our proposed learning objective enforces the model to learn environment-invariant representations against distribution shifts.

**Superiority to other OOD generalization methods.** Table 2 summarizes the results in comparsion with six state-of-the-art methods tailored for OOD learning, where we obtain the following observa-

Table 2: Evaluation with other OOD generalization methods on 6 out-of-distribution datasets from DrugOOD [27] in terms of ROC-AUC (%). The best and the runner-up in each columns are highlighted in **bolded** and underlined respectively. Note the baselines except ERM and MixUp all require environment labels. All methods including ours use GIN [63] as backbones.

| Dataset | IC50 | | | EC50 | | |
|---|---|---|---|---|---|---|
| Environment | **Assay** | **Scaffold** | **Size** | **Assay** | **Scaffold** | **Size** |
| **ERM** [56] | $70.93 \pm 2.10$ | $67.31 \pm 1.72$ | $67.40 \pm 0.56$ | $69.35 \pm 7.38$ | $63.92 \pm 2.09$ | $60.94 \pm 1.95$ |
| **IRM** [3] | $70.85 \pm 2.41$ | $66.06 \pm 1.23$ | $58.46 \pm 2.11$ | $69.94 \pm 1.03$ | $63.74 \pm 2.15$ | $58.30 \pm 1.51$ |
| **DeepCoral** [55] | $69.82 \pm 4.23$ | $66.36 \pm 2.57$ | $59.21 \pm 2.09$ | $69.42 \pm 3.35$ | $63.66 \pm 1.87$ | $56.13 \pm 1.77$ |
| **DANN** [18] | $70.00 \pm 1.03$ | $63.61 \pm 2.32$ | $65.77 \pm 0.47$ | $66.97 \pm 7.19$ | $64.33 \pm 1.82$ | $61.11 \pm 0.64$ |
| **MixUp** [70] | $70.22 \pm 3.66$ | $66.43 \pm 1.08$ | **$67.77 \pm 0.23$** | $70.62 \pm 2.12$ | $64.53 \pm 1.66$ | $62.67 \pm 1.41$ |
| **GroupDro** [50] | $69.98 \pm 1.74$ | $64.09 \pm 2.05$ | $58.46 \pm 2.69$ | $70.52 \pm 3.38$ | $64.13 \pm 1.81$ | $59.06 \pm 1.50$ |
| **Ours.** | **$71.38 \pm 0.68$** | **$68.02 \pm 0.55$** | $66.51 \pm 0.55$ | **$73.25 \pm 1.24$** | **$66.69 \pm 0.34$** | **$65.09 \pm 0.90$** |

Table 3: Ablation study on EC50-∗ by ROC-AUC (%). We show the results of MixUp that performs best among baselines on all EC50-∗ datasets and the naive ERM, which minimizes the average empirical loss on training data, for comparison. Notice that ERM and MixUp don't require manual specified environments labels. We also present the results of DANN, which requires manual specifications of environment and obtains competitive results with MixUp. All methods use GIN [63] as backbone.

| Method | **Assay** | **Scaffold** | **Size** |
|---|---|---|---|
| **ERM** (GIN + ERM loss) | $69.35 \pm 7.38$ | $63.92 \pm 2.09$ | $60.94 \pm 1.95$ |
| **MixUp** | $70.62 \pm 2.12$ | $64.53 \pm 1.66$ | $62.67 \pm 1.41$ |
| **DANN** | $66.97 \pm 7.19$ | $64.33 \pm 1.82$ | $61.11 \pm 0.64$ |
| Our architecture + ERM loss | $71.44 \pm 2.02$ | $65.99 \pm 0.42$ | $64.23 \pm 0.71$ |
| GIN + new learning objective | $72.07 \pm 1.14$ | $66.33 \pm 1.38$ | $64.43 \pm 1.10$ |
| DANN using our inferred environment label | $68.83 \pm 2.44$ | $64.95 \pm 1.07$ | $62.56 \pm 1.54$ |
| Our model using given environment label | $71.94 \pm 2.77$ | $66.29 \pm 0.85$ | $63.38 \pm 1.20$ |
| **Our full model** | **$73.25 \pm 1.24$** | **$66.69 \pm 0.34$** | **$65.09 \pm 0.90$** |

tions. Except on IC50-size, our method outperforms all baselines across all datasets due to its ability to enforce the molecule encoder to leverage environment-invariant substructures that more stably relate with the labels across environments. Our method ranks the third on IC50-size after MixUp and ERM. Different from the other methods, MixUp constructs more training exmaples and uses more data to train the model. That's why MixUp obtains best performance among all methods on IC50-size in our analysis. As for ERM, [27, 16] have pointed out that simple ERM shows better performance compared to subsequent OOD methods when datasets have relatively large environment counts. Even though we have set the environment number $k$ to a smaller value than the ground-truth number given by the dataset, we still need to prevent $k$ from being too small (see discussion in Sec. 4.4), leading to our poorer performance than ERM on IC50-size.

### 4.3   Ablation Study of Components

We analyze the contributions of different model components to the final performance in this section. Table 3 reports detailed ablation experimental results on EC50-assay, EC50-scaffold and EC50-size.

**Attention-based architecture.** We study the impact of the attention-based architecture introduced in Sec. 3.3 by assembling this architecture with ERM loss. We beat ERM and MixUp only with this architecture on three datasets. The results show that learning a representation for each substructure and then attentively aggregating these learned representations to obtain a final substructure-aware representation performs better than learning a representation for a complete molecular graph directly. This verifies our assumption that the substructure perspective is of importance to boosting performance of existing MRL methods. With the aid of such a substructure-grained learning architecture, the impact of our learning objective can be further strengthened.

**New learning objective.** To evaluate the impact of our proposed new learning objective, we equip GIN with this new objective. We can see compared to using the substructure-grained learning architecture only, only using the proposed new learning objective can bring more significant improvement. Thus, we can attribute the main superiority of our full model to this new objective. Combined with

the architecture discussed above, the new objective is able to better guide the molecule encoder to learn environment-invariant molecular representations against distribution shifts.

**Environment inference.** Now we turn to investigate the performance with respect to our proposed environment-inference module. One motivation for this module is that in reality manual specifications of environments may be unavailable due to the high price for labeling environments by experts. But when environment labels are available, how will be performance be like if directly utilizing the given environment partition? An ablation study is targeted on this. Taking the EC50-assay dataset as an example, it has given the environment partition and it specifies 47 environments in total. We utilize the given environment partition directly and keep the remaining parts in line with our full model. The results show that utilizing the given environment label, our method still can beat ERM and MixUp. But compared to our full model where we set the environment number $k$ to 20, it obtains inferior performance. Additionally, to further examine the effectiveness of our proposed environment inference method, we relabel the environment for each molecule for DANN according to our inferred environment partition. We can see that based on the new environment partition, DANN obtains better performance than using the initial given environment labels across three datasets. The reason why inferring environment instead can outperform directly using the given environment label is mainly due to the existing given partitions are often handcraftedrule-based and not structured. In contrast, letting the model learn a environment partition by itself may be more effective to some degree.

### 4.4 Hyper-parameter Sensitivity Study

We investigate the sensitivity of our method to these two hyper-parameters: the specified number of environments $k$, the trading-off parameter $\beta$ in Eq. 7. Fig. 3(a) shows the performance regarding different environment number $k$. It shows that the performance of our methods degrades when $k$ is too small (e.g. $k = 1, 5$) or too large (e.g. $k = 15$). When $k = 1$ i.e. we regard all training data as from only one environment, the performance is the poorest. This justifies that partitioning the training samples into different environments is necessary. Fig. 3(b) shows the results of our method by varying the trade-off parameter $\beta$. Our method obtains the worst performance when $\beta = 0$. This is mainly because Eq. 7 is reduced to the first term when $\beta = 0$. According to Theorem 2, without the second term of Eq. 7, the sufficiency condition of invariance principle cannot be satisfied, resulting in the performance degradation.

### 4.5 Risk Dynamics

Additionally, to shed insights of the ability of our method to lower the risks of different environments, we visualize the risk dynamic curve of some environments in Fig. 3(c). As is shown in Fig. 3(c), the difficulties of decreasing the risk on different environments are different. Though the risks of some environments vibrate violently at the beginning of training process (e.g. Env 1 and Env 2), with time elapsing, risks on all environments can decrease stably.

## 5 Conclusion

We have proposed a general framework which can incorporate any existing MRL method as backbone to improve their generalization ability against distribution shifts. Specifically, we devise a new learning scheme with its equivalent practical instantiation. We also develop an environment inference model to identify each molecule's corresponding environment without need of manual specifications of environments. Extensive experimental results on ten datasets demonstrate that our model yields consistent and significant improvements over various existing MRL methods as backbones. Additionally, our model achieves competitive or even superior performance compared to state-of-the-art models designed for OOD learning that require manual specified environment labels as extra inputs.

## Acknowledgement

This work was partly supported by National Key Research and Development Program of China (2020AAA0107600), National Natural Science Foundation of China (61972250, 72061127003), and Shanghai Municipal Science and Technology (Major) Project (22511105100, 2021SHZDZX0102).

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
