# A Proof for Proposition 1

*Proof.* Our goal is to minimize the Kullback-Leibler (KL) divergence between $q_\kappa(\mathbf{e}|\mathbf{G}, \mathbf{y})$ and $p_\tau(\mathbf{e}|\mathbf{G}, \mathbf{y})$. For the observed molecule graph and corresponding label tuple $(G, y)$,

$$
\begin{aligned}
& D_{KL}\left(q_\kappa(e|G, y) \| p_\tau(e|G, y)\right) \\
&= \int_{q_\kappa} q_\kappa(e|G, y) \log \frac{q_\kappa(e|G, y)}{p_\tau(e|G, y)} \mathrm{d}e = \int_{q_\kappa} q_\kappa(e|G, y) \log \frac{q_\kappa(e|G, y)p_\tau(y|G)p_\tau(G)}{p_\tau(e, G, y)} \mathrm{d}e \\
&= \left( \int_{q_\kappa} q_\kappa(e|G, y) \log q_\kappa(e|G, y) \mathrm{d}e + \int_{q_\kappa} p_\kappa(e|G, y) \log p_\tau(G) \mathrm{d}e \right. \\
&\quad \left. - \int_{q_\kappa} \log p_\tau(e, G, y) \mathrm{d}e \right) + \int_{q_\kappa} q_\kappa(e|G, y) \log p_\tau(y|G) \mathrm{d}e \\
&= \int_{q_\kappa} q_\kappa(e|G, y) \log \frac{q_\kappa(e|G, y)}{p_\tau(y|G, e)p_\tau(e|G)} \mathrm{d}e + \log p_\tau(y|G) \\
&= \mathbb{E}_{q_\kappa}[\log q_\kappa(e|G, y) - \log p_\tau(y|G, z) - \log p_\tau(e|G)] + \log p_\tau(y|G) \\
&= -\mathbb{E}_{q_\kappa}[\log p_\tau(y|G, e)] + \mathbb{E}_{q_\kappa}[\log q_\kappa(e|G, y) - \log p_\tau(e|G)] + \log p_\tau(y|G) \\
&= -\underbrace{\left(\mathbb{E}_{q_\kappa}[\log p_\tau(y|G, e)] - D_{KL}(q_\kappa(e|G, y) \| p_\tau(e|G))\right)}_{\mathcal{L}(\tau,\kappa;G,y)} + \log p_\tau(y|G) \\
&= -\mathcal{L}(\tau, \kappa; G, y) + \log p_\tau(y|G)
\end{aligned}
\tag{8}
$$

Rearrange Eq. 8 and we can get,

$$
\mathcal{L}(\tau, \kappa; G, y) = -D_{KL}\left(q_\kappa(e|G, y) \| p_\tau(e|G, y)\right) + \log p_\tau(y|G). \tag{9}
$$

The defined $\mathcal{L}(\tau, \kappa; G, y)$ is called *Evidence Lower BOund* (ELBO) [22]. According to Eq. 9, maximizing this ELBO is equivalent to minimizing the KL divergence and maximizing $\log p_\tau(y|G)$. For the observed molecule graph and corresponding label tuple $(G, y)$, we obtain the ELBO:

$$
\mathcal{L}(\tau, \kappa; G, y) = \mathbb{E}_{q_\kappa}[\log p_\tau(y|G, e)] - D_{KL}(q_\kappa(e|G, y) \| p(e|G)). \tag{10}
$$

We thus conclude the proof. $\qquad\square$

# B Proofs for Theorems

In this paper, we extend the invariance assumption [49, 3] to molecule representation learning:

**Assumption 1.** *Given a molecular graph $G$, there exists an encoder $\Phi$ yielding a graph-level representation $r_G \in \mathbb{R}^d$. Define $\mathbf{r}$ as a random variable of $r_G$ and it satisfies: 1) (Invariance condition): $p(\mathbf{y}|\mathbf{r}, \mathbf{e}) = p(\mathbf{y}|\mathbf{r})$, and 2) (Sufficiency condition): $\mathbf{y} = h(\mathbf{r}) + \mathbf{n}$, where $h$ is a non-linear function, $\mathbf{n}$ is a independent noise.*

With the terminology of information theory, we present a useful lemma [61] that interprets the invariance and sufficiency conditions in Assumption 1:

**Lemma 1.** *In terms of information theory, the two conditions in Assumption 1 can be equivalently expressed as 1) invariance: $\mathrm{I}(\mathbf{y}; \mathbf{e}|\mathbf{r}) = 0$ and 2) sufficiency: $\mathrm{I}(\mathbf{y}; \mathbf{r})$ is maximized.*

*Proof.* For the invariance, it can be obtained by the fact that

$$
\mathrm{I}(\mathbf{y}; \mathbf{e}|\mathbf{r}) = \mathbb{E}_{p(\mathbf{e}, \mathbf{r})}[D_{KL}(p(\mathbf{y}|\mathbf{e}, \mathbf{r}) \| p(\mathbf{y}|\mathbf{r}))] \tag{11}
$$

For the sufficiency, we first prove that every triplet $(\mathbf{G}, \mathbf{r}, \mathbf{y})$ satisfying that $\mathbf{y} = h(\mathbf{r}) + \mathbf{n}$ would also satisfy $\mathbf{r} = \arg\max_\mathbf{r} \mathrm{I}(\mathbf{y}; \mathbf{r})$. We prove it by contradiction. Assume that $\mathbf{r} \neq \arg\max_\mathbf{r} \mathrm{I}(\mathbf{y}; \mathbf{r})$ and there exists $\mathbf{r}'$ with $\mathbf{r}' = \arg\max_\mathbf{r} \mathrm{I}(\mathbf{y}; \mathbf{r})$ with $\mathbf{r} \neq \mathbf{r}'$. Then there exists another random variable $\tilde{\mathbf{r}}$ and a mapping function $f_m$ such that $\mathbf{r}' = f_m(\mathbf{r}, \tilde{\mathbf{r}})$. Then we will have $\mathrm{I}(\mathbf{y}; \mathbf{r}') = \mathrm{I}(\mathbf{y}; \mathbf{r}, \tilde{\mathbf{r}}) = \mathrm{I}(h(\mathbf{r}); \mathbf{r}, \tilde{\mathbf{r}}) = \mathrm{I}(h(\mathbf{r}); \mathbf{r}) = \mathrm{I}(\mathbf{y}; \mathbf{r})$, which leads to contradiction.

Then we prove that every triplet $(\mathbf{G}, \mathbf{r}, \mathbf{y})$ satisfying that $\mathbf{r} = \arg\max_\mathbf{r} \mathrm{I}(\mathbf{y}; \mathbf{r})$ would also satisfy $\mathbf{y} = h(\mathbf{r}) + \mathbf{n}$ by contradiction. Suppose that $\mathbf{y} \neq h(\mathbf{r}) + \mathbf{n}$ and there exists $\mathbf{r}' \neq \mathbf{r}$ with $\mathbf{y} = h(\mathbf{r}') + \mathbf{n}$. Then the inequality $\mathrm{I}(h(\mathbf{r}'); \mathbf{r}) < \mathrm{I}(h(\mathbf{r}'); \mathbf{r}')$ holds. That means $\mathbf{r}' = \arg\max_r \mathrm{I}(\mathbf{y}; \mathbf{r})$, leading to contradiction. $\qquad\square$

## B.1 Proof for Theorem 1

*Proof.* According to the dependency relationship $\mathbf{z} \leftarrow \mathbf{G} \rightarrow \mathbf{y}$, we have

$$
\begin{aligned}
&\mathrm{I}(\mathbf{y}; \mathbf{e}|\mathbf{z}) \\
&= D_{KL}(p(\mathbf{y}|\mathbf{z}, \mathbf{e}) \parallel p(\mathbf{y}|\mathbf{z})) \\
&= D_{KL}(p(\mathbf{y}|\mathbf{z}, \mathbf{e}) \parallel \mathbb{E}_{p(\mathbf{e}|\mathbf{G})}[p(\mathbf{y}|\mathbf{z}, \mathbf{e})]) \\
&= D_{KL}\left(q(\mathbf{y}|\mathbf{z}) \parallel \mathbb{E}_{p(\mathbf{e}|\mathbf{G})}[p(\mathbf{y}|\mathbf{G}, \mathbf{e})]\right) - D_{KL}\left(q(\mathbf{y}|\mathbf{z}) \parallel p(\mathbf{y}|\mathbf{z}, \mathbf{e})\right) \\
&\quad - D_{KL}\left(\mathbb{E}_{p(\mathbf{e}|\mathbf{G})}[p(\mathbf{y}|\mathbf{z}, \mathbf{e})] \parallel \mathbb{E}_{p(\mathbf{e}|\mathbf{G})}[p(\mathbf{y}|\mathbf{G}, \mathbf{e})]\right) \\
&\leq D_{KL}\left(q(\mathbf{y}|\mathbf{z}) \parallel \mathbb{E}_{p(\mathbf{e}|\mathbf{G})}[p(\mathbf{y}|\mathbf{G}, \mathbf{e})]\right).
\end{aligned} \tag{12}
$$

Next, we have

$$
\begin{aligned}
&D_{KL}\left(q(\mathbf{y}|\mathbf{z}) \parallel \mathbb{E}_{p(\mathbf{e}|\mathbf{G})}[p(\mathbf{y}|\mathbf{G}, \mathbf{e})]\right) \\
&= \mathbb{E}_{G \sim p(\mathbf{G})} \mathbb{E}_{y_G \sim p(\mathbf{y}|\mathbf{G}=G)} \mathbb{E}_{z_G \sim q(\mathbf{z}|\mathbf{G}=G)} \left[\log \frac{q(\mathbf{y}=y_G|\mathbf{z}=z_G)}{\mathbb{E}_{p(\mathbf{e}|\mathbf{G})}[p(\mathbf{y}=y_G|\mathbf{G}=G, \mathbf{e}=e)]}\right] \\
&= \frac{1}{|\mathcal{G}|} \sum_{(G, y_G) \in \mathcal{G}} \mathbb{E}_{z_G \sim q(\mathbf{z}|\mathbf{G}=G)} \left[\log \frac{q(\mathbf{y}=y_G|\mathbf{z}=z_G)}{\mathbb{E}_{p(\mathbf{e}|\mathbf{G})}[p(\mathbf{y}=y_G|\mathbf{G}=G, \mathbf{e}=e)]}\right].
\end{aligned} \tag{13}
$$

Based on Jensen Inequality and Triangle Inequality, we can obtain that $D_{KL}\left(q(\mathbf{y}|\mathbf{z}) \parallel \mathbb{E}_{p(\mathbf{e}|\mathbf{G})}[p(\mathbf{y}|\mathbf{G}, \mathbf{e})]\right)$ is upper bounded by:

$$
\frac{1}{|\mathcal{G}|} \sum_{(G, y) \in \mathcal{G}} \left|\log q_\theta(y|G) - \mathbb{E}_{p(\mathbf{e}|\mathbf{G})}[\log p_\tau(y|G, e)]\right|. \tag{14}
$$

Thus we can prove that minimizing term ① in Eq. 7 is equivalent to $\min_{q_\theta(\mathbf{y}|\mathbf{z}), q_\theta(\mathbf{z}|\mathbf{G})} \mathrm{I}(\mathbf{y}; \mathbf{e}|\mathbf{z})$.
$\square$

## B.2 Proof for Theorem 2

*Proof.* Given the dependency relationship $\mathbf{z} \leftarrow \mathbf{G} \rightarrow \mathbf{y}$, we hold $\max_{q(\mathbf{z}|\mathbf{G})} \mathrm{I}(\mathbf{y}; \mathbf{z})$ is equivalent to $\min_{q(\mathbf{z}|\mathbf{G})} \mathrm{I}(\mathbf{y}; \mathbf{G}|\mathbf{z})$. Also we have

$$
\begin{aligned}
\mathrm{I}(\mathbf{y}; \mathbf{G}|\mathbf{z}) &= D_{KL}(p(\mathbf{y}|\mathbf{G}, \mathbf{e}) \| p(\mathbf{y}|\mathbf{z}, \mathbf{e})) \\
&= D_{KL}(p(\mathbf{y}|\mathbf{G}, \mathbf{e}) \| q(\mathbf{y}|\mathbf{z})) - D_{KL}(p(\mathbf{y}|\mathbf{z}, \mathbf{e}) \| q(\mathbf{y}|\mathbf{z})) \\
&\leq D_{KL}(p(\mathbf{y}|\mathbf{G}, \mathbf{e}) \| q(\mathbf{y}|\mathbf{z})),
\end{aligned} \tag{15}
$$

Based on this, we will have

$$
\mathrm{I}(\mathbf{y}; \mathbf{G}|\mathbf{z}) \leq \min_{q(\mathbf{y}|\mathbf{z})} D_{KL}(p(\mathbf{y}|\mathbf{G}, \mathbf{e}) \| q(\mathbf{y}|\mathbf{z})). \tag{16}
$$

Then we can also derive the following inequality via Jensen Inequality:

$$
\begin{aligned}
D_{KL}(p(\mathbf{y}|\mathbf{G}, \mathbf{e}) \| q(\mathbf{y}|\mathbf{z})) &= \mathbb{E}_{\mathbf{e}} \mathbb{E}_{G \sim p_e(\mathbf{G})} \left[\mathbb{E}_{y_G \sim p_e(\mathbf{y}|\mathbf{G}=G)} \mathbb{E}_{z \sim q(\mathbf{z}|\mathbf{G}=G)} \left[\log \frac{p_e(\mathbf{y}=y_G|\mathbf{G}=G)}{q(\mathbf{y}=y_G|\mathbf{z}=z_G)}\right]\right] \\
&\leq \mathbb{E}_{\mathbf{e}} \left[\frac{1}{|\mathcal{G}^e|} \sum_{(G, y_G) \in \mathcal{G}^e} \log \frac{p_e(\mathbf{y}=y_G|\mathbf{G}=G)}{\mathbb{E}_{z_G \sim q(\mathbf{z}|\mathbf{G}=G)} q(\mathbf{y}=y_G|\mathbf{z}=z_G)}\right] \\
&= C + \mathbb{E}_{\mathbf{e}} \left[-\frac{1}{|\mathcal{G}^e|} \sum_{(G, y_G) \in \mathcal{G}^e} \log q(\mathbf{y}=y_G|\mathbf{G}=G)\right],
\end{aligned} \tag{17}
$$

where $C$ is a constant. Then the problem $\min_{q(\mathbf{y}|\mathbf{z})} D_{KL}(p(\mathbf{y}|\mathbf{G}, \mathbf{e}) \| q(\mathbf{y}|\mathbf{z}))$ can be solve by

$$
\min \mathbb{E}_{\mathbf{e}} \left[\frac{1}{|\mathcal{G}^e|} \sum_{(G, y_G) \in \mathcal{G}^e} [-\log q_\theta(\mathbf{y}=y_G|\mathbf{G}=G)]\right], \tag{18}
$$

which means minimizing term ② in Eq. 7 contributes to $\max_{q_\theta(\mathbf{y}|\mathbf{z}), q_\theta(\mathbf{z}|\mathbf{G})} \mathrm{I}(\mathbf{z}; \mathbf{y})$.
$\square$

## C Implementation Details

### C.1 Baselines

This section describes training configurations for all baselines, which are compared in this paper.

**Three backbones.** We adapt three backbones into our method, namely, **GCN** [30], **Graph-SAGE** [21] and **GIN** [63]. We also emphasize the comparison with their augmented versions, i.e. "**+ virtual node**" [19, 25, 38]. For GCN and GIN, we use the implementations provided by Open Graph Benchmark [23][5]. We implement GraphSAGE and its corresponding augmented version by ourselves. For these baselines, grid search of learning rate over $\{1e-2, 5e-3, 1e-3, 5e-4, 1e-4\}$ and dropout rate over $\{0.1, 0.3, 0.5\}$ is performed to select the best parameters. The embedding size of all methods including ours are all set to 256 for the sake of fairness.

- **GCN** [30] is a scalable approach on graph-structured data that is based on an efficient variant of convolutional neural networks.

- **GIN** [63] generalizes the Weisfeiler-Lehman (WL) graph isomorphism test [36] and hence achieves maximum discriminative power among GNNS.

- **GraphSAGE** [21] learns a function that generates embeddings by sampling and aggregating features from a node's local neighborhood.

- **GCN/GIN/GraphSAGE + virtual node** [19, 25, 38] is a variant of the original method augmented by an additional node connecting to all nodes in the raw graph.

**Models tailored for OOD learning.** We compare our method against six state-of-the-art methods: **ERM** [56], **IRM** [3], **DeepCoral** [55], **DANN** [18], **MixUp** [70] and **GroupDro** [50]. We use the implementations of these six method provided by DrugOOD[6]. We search for the optimal hyper-parameters by ranging learning rate over $\{1e-3, 5e-4, 1e-4, 5e-5, 1e-5\}$ and dropout rate over $\{0.1, 0.3, 0.5\}$. The embedding size of all models including ours are all set to 128 for fairness.

- **ERM** [56] minimizes the average empirical loss on training data.

- **IRM** [3] penalizes feature distributions for environments that have different optimum predictors. We set the penalty weight and the penalty anneal iteration to 10 and 500, respectively.

- **DeepCoral** [55] penalizes differences in the means and covariances of the feature distributions for each environment, which are exactly the distribution of last layer activations in a neural network. The penalty weight is set to $0.001$.

- **DANN** [18] encourages feature representations to be consistent across domains. We set to the inverse factor to $0.2$.

- **MixUp** [70] constructs additional virtual samples for training from two examples which are randomly sampled from the training data. We set the probability and interpolate strength to $0.1$.

- **GroupDro** [50] minimizes the worst-case training loss over a set of pre-defined environments. The step size is set to $0.001$.

### C.2 Our Method

We implement our method in Pytorch. As for experiments on OGB datasets, we implement the Environment Classifier, the Conditional GNN and the Substructure Encoder which are mentioned in Sec. 3.3 all in Graph Isomorphism Network (GIN) [63]. We use grid search on validation set for hyper-parameter tuning by ranging learning rate from $\{1e-2, 5e-3, 1e-3, 5e-4, 1e-4, 5e-5, 1e-5\}$, dropout rate from $\{0.1, 0.2, 0.3, 0.4, 0.5\}$, the trading-off parameter $\beta$ from $\{0.5, 1, 2, 4\}$, the environment count $k$ from $\{5, 10, 15, 20, 40, 80\}$. As for the prior $p(\mathbf{e}|\mathbf{G})$, we set it to a *Uniform* distribution or a discrete *Gaussian* distribution. We use CrossEntropyLoss for all models and the Adam optimizer is used for gradient-based optimization.

---

[5] https://github.com/snap-stanford/ogb
[6] https://github.com/tencent-ailab/DrugOOD

# D More Details of Datasets

In this paper, we use ten publicly available benchmark datasets in total. Four of them, namely, BACE, BBBP, SIDER and HIV are released by Open Graph Benchmark (OGB) [23]. The rest six are released by DrugOOD [27], i.e. IC50-assay, IC50-scaffold, IC50-size, EC50-assay, EC50-scaffold and EC50-size. We provide detailed descriptions for them as below.

- **BBBP** is a dataset of Brain-Blood Barrier Penetration. Each molecule has a label indicating whether it can penetrate through brain cell membrane to enter central nervous system.
- **BACE** is a dataset of binding affinity against human beta-secretas 1. Each molecule has a label indicating whether it binds to human beta-secretase 1.
- **SIDER** is a dataset of marked drugs and adverse drug reactions (ADRs). Molecules are grouped into 27 system organ classes.
- **HIV** is a dataset of HIV antiviral activity. Each molecule has an active or inactive label.
- **IC50/EC50-scaffold/assay/size** are datasets generated by the automated dataset curator provided by DrugOOD from the large-scale bioassay deposition website ChEMBL [42]. The suffix specifies the splitting scheme. These six datasets target on ligand-based affinity prediction (LBAP). Each molecule has an active or inactive label.

Notice that the phenomenon that there exist a few invariant substructures w.r.t. certain property indeed exists in the datasets we use in our paper. Taking **HIV** dataset as an example, *salicylhydrazide* substructure displays potent HIV-1 integrase (IN) inhibitory activity, which has been identified by previous studies [1, 45]. Additionally, for **BBBP** dataset, as pointed out by recent studies [52, 53], some substructures are closely related to brain-blood barrier penetration.

All these ten datasets do not contain personally identifiable information or offensive content. Table 4 shows the detailed statistics of datasets. For all datasets, we adopt the default training-validation-test split as shown in Table 4. We use all molecules in the training set to optimize the model parameters. Then, we select hyper-parameters using the validation set, and we report the results on test molecule set for the model that achieves the best results on the validation set.

Table 4: **Summary of datasets used in this paper.** #Train/#Valid/#Test denotes the number of samples in the training/validation/test set, respectively. #Total is the sum of #Train, #Valid and #Test. #Tasks is the output dimensionality required for prediction. Additionally, we also list which split scheme is adopted and whether the manual specification of environments is available for each dataset.

| | Dataset | #Train | #Valid | #Test | #Total | #Tasks | Split Scheme | Specify Environments? |
|---|---|---|---|---|---|---|---|---|
| **OGB** | BACE | 1,210 | 151 | 152 | 1,513 | 1 | Scaffold | ✗ |
| | BBBP | 1,631 | 204 | 204 | 2,039 | 1 | Scaffold | ✗ |
| | SIDER | 1,141 | 143 | 143 | 1,427 | 27 | Scaffold | ✗ |
| | HIV | 32,901 | 4,113 | 4,113 | 41,127 | 1 | Scaffold | ✗ |
| **DrugOOD** | EC50-assay | 4,540 | 2,572 | 2,490 | 9,602 | 1 | Assay | ✓ |
| | EC50-scaffold | 2,570 | 2,532 | 2,533 | 7,635 | 1 | Scaffold | ✓ |
| | EC50-size | 4,684 | 2,313 | 2,398 | 9,395 | 1 | Size | ✓ |
| | IC50-assay | 34,179 | 19,028 | 19,028 | 72,235 | 1 | Assay | ✓ |
| | IC50-scaffold | 21,519 | 19,041 | 19,048 | 59,608 | 1 | Scaffold | ✓ |
| | IC50-size | 36,597 | 17,660 | 16,415 | 70,672 | 1 | Size | ✓ |

Table 5: We count the number of scaffolds that contain 1, 2, 3, 4 and 5 samples, respectively.

| Size | Number |
|------|--------|
| 1 | 14, 295 |
| 2 | 2, 330 |
| 3 | 862 |
| 4 | 449 |
| 5 | 255 |

Next, let's discuss on the details of HIV dataset, which is released by Open Graph Benchmark (OGB) [23]. OGB adopts scaffold splitting scheme to split the HIV into train/validation/test set. We count the number of scaffolds that only contain 1, 2, 3, 4 and 5 molecules, respectively, and summarize the statistics in Table 5. Notice that HIV has $19,076$ scaffolds in total. We can see there are $18,191$ scaffolds containing less or equal to $5$ molecules, accounting for $95.45\%$ of the total scaffold count. HIV has a great deal of environments that contains few samples, which poses great challenge to directly applying some existing OOD generalization methods to datasets like HIV [27]. Thus, for datasets released by OGB, partitioning the mocecules into different environments according to their scaffolds may not be suitable in practice. Such a observation motivates us to propose the environment-inference model.

## E    Understanding the Data-generating Process

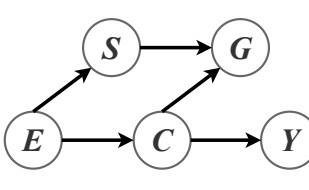

Figure 4: SCM

We provide a causal perspective to understand the data-generating process. Recalling the two molecules *Cyclopropanol* ($C_3H_6O$) and *1,4-Cyclohexanediol* ($C_6H_{12}O_2$) used for illustration in Sec. 1, they are sampled from different environments. Because both of them contain the hydroxy ($-OH$), which we can call invariant or causal substructure in this case, these two molecules are readily soluble in water. We formalize such a date-generating process of molecule property prediction with a general Structural Causal Model (SCM) [20, 46] in Fig. 4. The abstract data variables are denoted by the nodes and the directed arrows represent the causalities. This SCM illustrates the causalities among variables: $E$ as the environment, $S$ as the spurious substructures, $C$ as the invariant/causal substructures w.r.t $Y$, $G$ as the instance molecule graph, $Y$ as the ground-truth label.

- $S \leftarrow E \rightarrow C$: the environmental variable impacts the underlying data generating distribution. Furthermore, substructures could be divided into causal and spurious ones across all environments.

- $S \rightarrow G \leftarrow C$: an instance molecule graph is made up of the causal and spurious substructures.

- $C \rightarrow Y$: $Y$, the ground-truth label, is only determined by $C$. This causation is the focus of our work.

Taking *Cyclopropanol* ($C_3H_6O$) as an example, we can specify $E$ as the *3C-ring* scaffold, $C$ as the substructure hydroxy ($-OH$), $S$ as the substructures aside from hydroxy, $G$ as *Cyclopropanol*, $Y$ as good water solubility. The good water solubility $Y$ is only attributed to the invariant substructure hydroxy, i.e. $C$, rather than other spurious substructures $S$.

Existing MRL methods do not differentiate invariant and spurious substructures. Hence, the spurious correlations between irrelevant substructures $S$ and the target label $Y$ will be encoded to learned molecular representations. When tested on unseen environments, the downstream classifier will be easily misled by these spurious correlations [3].

## F    Notations

We summarize the notations used in this paper in Table 6.

## G    Sensitivity to Molecule Segmentation Method

For all experiments in our original paper, we all adopt *breaking retrosynthetically interesting chemical substructures* (BRICS) to segment molecule into substructures, which is widely used in other works related to molecules, e.g., [60, 11]. To investigate the sensitivity of our method to different decomposing strategies, we adopt another molecule segmentation method called *retrosynthetic combinatorial analysis procedure* (RECAP) [37], which is also available as an API in RDKit package. RECAP and BRICS decompose molecules based on two different rules. We conduct experiments on EC50-assay/scaffold/size three datasets and the comparisons are summarized in Table 7. We can

Table 6: Notations.

| Notation | Description |
|---|---|
| $e$ | an environment instance |
| $\mathbf{e}$ | a random variable of $e$ |
| $\mathcal{E}$ | the support of environments |
| $l(\cdot)$ | the loss function |
| $\mathcal{R}(\cdot)$ | the risk function |
| $G$ | a molecular graph instance |
| $\mathbf{G}$ | a random variable of $G$ |
| $y$ | a ground-truth label instance |
| $\mathbf{y}$ | a random variable of $y$ |
| $\mathcal{G}$ | a dataset set, i.e. $\{(G, y)\}$ |
| $\psi$ | the environment-inference model |
| $\Phi$ | the molecule encoder |
| $\omega$ | the final predictor |
| $\mathbf{z}$ | the denotation of $\Phi(\mathbf{G})$ |
| $f$ | $\omega \circ \Phi$ |
| $\kappa$ | the learnable parameters of the Environment Classifier |
| $\tau$ | the learnable parameters of the Conditional GNN |
| $\theta$ | the learnable parameters of $\Phi$ and $\omega$ |
| $k$ | hyper-parameter: the environment count |
| $\beta$ | hyper-parameter: the trading-off parameter in Eq. 7 |

Table 7: Comparisons on 3 out-of-distribution datasets in terms of ROC-AUC (%). The best and the runner-up in each columns are highlighted in **bolded** and underlined respectively. Note the baselines except ERM and MixUp all require environment labels. All methods including ours use GIN [63] as backbones. Each experiment is repeated 5 times with mean and standard deviation reported.

| Dataset | EC50 | | |
|---|---|---|---|
| Environment | **Assay** | **Scaffold** | **Size** |
| **ERM** [56] | $69.35 \pm 7.38$ | $63.92 \pm 2.09$ | $60.94 \pm 1.95$ |
| **IRM** [3] | $69.94 \pm 1.03$ | $63.74 \pm 2.15$ | $58.30 \pm 1.51$ |
| **DeepCoral** [55] | $69.42 \pm 3.35$ | $63.66 \pm 1.87$ | $56.13 \pm 1.77$ |
| **DANN** [18] | $66.97 \pm 7.19$ | $64.33 \pm 1.82$ | $61.11 \pm 0.64$ |
| **MixUp** [70] | $70.62 \pm 2.12$ | $64.53 \pm 1.66$ | $62.67 \pm 1.41$ |
| **GroupDro** [50] | $70.52 \pm 3.38$ | $64.13 \pm 1.81$ | $59.06 \pm 1.50$ |
| **Ours + RECAP** | $\underline{72.72 \pm 3.94}$ | $66.34 \pm 0.52$ | $\mathbf{65.48 \pm 1.10}$ |
| **Ours + BRICS** | $\mathbf{73.25 \pm 1.24}$ | $\mathbf{66.69 \pm 0.34}$ | $\underline{65.09 \pm 0.90}$ |

see that RECAP and BRICS show competitive performance on our model and both outperform the baselines by large margins.

# H   Future Direction

Sometimes, bio-chemical properties are affected by interactions between substructures. To encode such interactions between substructures into the final learned molecular representation, we utilize the permutation equivariant Set Attention Block (SAB) proposed in Set Transformer [35]. SAB takes a representation set of any size as input and outputs a representation set of equal size. SAB is able to encode pairwise and higher-order interactions between elements in input sets into outputs. We add such a SAB after the Substructure Encoder. For each molecule, we feed the represenions of its substructures to SAB to obtain new substruture representations. In this way, the final molecule representation could model interactions between substructures. We conduct experiments on EC50-assay/scaffold/size to examine the performance of adding such a SAB. As demonstrated in Table 8,

Table 8: Comparisons on 3 out-of-distribution datasets in terms of ROC-AUC (%). The best and the runner-up in each columns are highlighted in **bolded** and underlined respectively. Note the baselines except ERM and MixUp all require environment labels. All methods including ours use GIN [63] as backbones. Each experiment is repeated 5 times with mean and standard deviation reported.

| Dataset | EC50 | | |
|---|---|---|---|
| Environment | **Assay** | **Scaffold** | **Size** |
| **ERM** [56] | $69.35 \pm 7.38$ | $63.92 \pm 2.09$ | $60.94 \pm 1.95$ |
| **IRM** [3] | $69.94 \pm 1.03$ | $63.74 \pm 2.15$ | $58.30 \pm 1.51$ |
| **DeepCoral** [55] | $69.42 \pm 3.35$ | $63.66 \pm 1.87$ | $56.13 \pm 1.77$ |
| **DANN** [18] | $66.97 \pm 7.19$ | $64.33 \pm 1.82$ | $61.11 \pm 0.64$ |
| **MixUp** [70] | $70.62 \pm 2.12$ | $64.53 \pm 1.66$ | $62.67 \pm 1.41$ |
| **GroupDro** [50] | $70.52 \pm 3.38$ | $64.13 \pm 1.81$ | $59.06 \pm 1.50$ |
| **Ours** | $\mathbf{73.25 \pm 1.24}$ | $\underline{66.69 \pm 0.34}$ | $\mathbf{65.09 \pm 0.90}$ |
| **Ours + SAB** | $\underline{73.15 \pm 2.69}$ | $\mathbf{67.26 \pm 1.54}$ | $\underline{64.83 \pm 1.07}$ |

we can see that adding such a SAB further improves our model on EC50-scaffold. This design is a naive attempt but brings us some valuable insights.

## I  Limitations

Some studies [27, 16] have empirically shown that existing models designed for OOD learning may fail to outperform the simple ERM [56] model when the environment count is large. Though we can relabel the environment for each molecule according to the new environment partition inferred by our devised environment-inference module, we still need to set the environment count $k$ to a relatively larger value than that of other OOD datasets from other domain, e.g. Camelyon17 [4], which only contains five environments. Thus, using our inferred environment partition, existing models designed for OOD learning might still be inferior to ERM in some cases.

## J  Potential Negative Impacts

As far as we are concerned, we have not identified any negative social impact of this work.