# OpenReview forum: "Learning Substructure Invariance for Out-of-Distribution Molecular Representations"
_NeurIPS.cc/2022/Conference — NeurIPS 2022 Accept_

### Official Review · Reviewer_WE4q · 2022-06-26

**Rating:** 7
**Confidence:** 5
**Soundness:** 4 excellent
**Presentation:** 4 excellent
**Contribution:** 3 good

**Summary:**

This paper is mainly designing a learning framework to learn causal-invariance molecular representations, improving the generalization of MRL in OOD settings. This paper observes that the molecule usually consists of two parts: one part is causal substructure that determines the behaviors of molecular properties, another part is spurious substructure that constitute the molecular structures but has no impact on molecular behaviors. Based on this observation, this paper builds a structural casual model to learn more robust molecular representation learning. Specifically, new model includes substructure-aware attention-based molecular encoders and VAE-based environment inference model. To enable two-stage optimization, this paper proposes a new learning objective function consisting of the ELBO objective and the ERM objective.

**Questions:**

(1) Does different decomposing strategy affect the model performance?
(2) What if the model is trained in a simple multi-task learning setting? In other words, setting environment prediction as an auxiliary task? Would this have equivalent performance to the proposed method? Why it is necessary to design environment prediction in a VAE learning way?

**Limitations:**

This reviewer thinks the proposed method in this paper has the major limitations that the scaffold and decomposing strategy may not be robust to OOD setting although adding this additional knowledge could improve robustness of MRL.

**Strengths And Weaknesses:**

Strengths:
(1) This reviewer thinks that this paper has good novelty. It proposes an important but overlooked observation in molecular representation learning. In short, the problem addressed in this paper is significant, the proposed solution is sufficiently novelty and the motivation is reasonable;
(2) The presentation of each section is quite clear and well-organized. It is not hard for this reviewer to understand the novel idea and elegant design of new MRL paradigm;
(3) Sufficient experiments over different public benchmarks under OOD settings and ablation studies demonstrate the effectiveness of each component;
Weaknesses:
(1) Decomposing strategies (BRICS) and scaffold introduction as environment actually implicitly introduce additional knowledge to MRL. The decomposing step and scaffold introduction both follow hand-engineered rules by domain experts and may have non-trivial impacts on model performance.

---

> ### Author Response · Authors · 2022-08-02
> **Response to Reviewer WE4q**
>
> Thank you for the valuable comments. We are glad that you found our work well written, and appreciated our elegant approach, reasonable motivation, theoretical support and solid experimental evaluation. Below we provide detailed responses.
>
> **Q1: Sensitivity of our model to different decomposing strategyies.**
>
> We supplement new experimental results. For all experiments in our original paper, we all adopt *breaking retrosynthetically interesting chemical substructures* (BRICS) to segment molecule into substructures, which is widely used in other works related to molecules [1,2,3]. To investigate the sensitivity of our method to different decomposing strategies, we adopt another molecule segmentation method called *retrosynthetic combinatorial analysis procedure* (RECAP)[4], which is also available as an API in RDKit package. RECAP and BRICS decompose molecules based on two different rules. Due to limited time, we only conduct experiments on EC50-assay/scaffold/size three datasets and the comparisions are summarized in table below. We can see that RECAP and BRICS show competitive performance on our model and both outperform the baselines by large margins.
>
> |                |       **EC50-assay**       |     **EC50-scaffold**      |       **EC50-size**        |
> |:-------------- |:--------------------------:|:--------------------------:|:--------------------------:|
> | **ERM**        |       $69.35\pm7.38$       |       $63.92\pm2.09$       |       $60.94\pm1.95$       |
> | **IRM**        |       $69.94\pm1.03$       |       $63.74\pm2.15$       |       $58.30\pm1.51$       |
> | **DeepCoral**  |       $69.42\pm3.35$       |       $63.66\pm1.87$       |       $56.13\pm1.77$       |
> | **DANN**       |       $66.97\pm7.19$       |       $64.33\pm1.82$       |       $61.11\pm0.64$       |
> | **MixUp**      |       $70.62\pm2.12$       |       $64.53\pm1.66$       |       $62.67\pm1.41$       |
> | **GroupDro**   |       $70.52\pm3.38$       |       $64.13\pm1.81$       |       $59.06\pm1.50$       |
> | **Ours-RECAP** | $\underline{72.72\pm3.94}$ | $\underline{66.34\pm0.52}$ |  $\mathbf{65.48\pm1.10}$   |
> | **Ours-BRICS** |  $\mathbf{73.25\pm1.24}$   |  $\mathbf{66.69\pm0.34}$   | $\underline{65.09\pm0.90}$ |
>
> **Q2: “What if the model is trained in a simple multi-task learning setting? In other words, setting environment prediction as an auxiliary task? Would this have equivalent performance to the proposed method? Why it is necessary to design environment prediction in a VAE learning way?”**
>
> This is an insightful question. But our method is a little bit different from multi-task learning. $\mathcal{L}\_{elbo}$  only influences the parameters of environment inference model  while $\mathcal{L}\_{inv}$ only influences the parameters of molecule encoder. Thus, we adopt a simple two-stage training strategy here. However, training the model in a multi-task-learning way can be a potiential direction, which we leave for future investigation. As mentioned in our paper, we wants to maximize the log-likelihood of $p_{\tau}(\mathbf{y}|\mathbf{G})$ and then obtain the posterior $p_{\tau}(\mathbf{e}\vert \mathbf{G},\mathbf{y})$, which are parameterized by $\tau$. Since there is no analytical solutions to the true posterior, we adopt variational inference (VI) to approximate it as an initial attempt and have proved the correctness of the objective in Eqn. 6 in Appendix A. There might exist alternative methods to realize environment inference, which we believe can be explored by future works.
>
> **Reference (all are new and will be added in the main paper):**
>
> [1] [Improving Molecular Contrastive Learning via Faulty Negative Mitigation and Decomposed Fragment Contrast.](https://pubs.acs.org/doi/pdf/10.1021/acs.jcim.2c00495)
>
> [2] [SafeDrug: Dual Molecular Graph Encoders for Recommending Effective and Safe Drug Combinations.](https://www.ijcai.org/proceedings/2021/0514.pdf)
>
> [3] [An Evolutionary Fragment-based Approach to Molecular Fingerprint Reconstruction.](https://dl.acm.org/doi/pdf/10.1145/3512290.3528824)
>
> [4] [RECAPRetrosynthetic Combinatorial Analysis Procedure:  A Powerful New Technique for Identifying Privileged Molecular Fragments with Useful Applications in Combinatorial Chemistry.](https://pubs.acs.org/doi/pdf/10.1021/ci970429i)

---

### Official Review · Reviewer_MgvC · 2022-07-01

**Rating:** 5
**Confidence:** 2
**Soundness:** 2 fair
**Presentation:** 2 fair
**Contribution:** 2 fair

**Summary:**

This work aims to address the problem of the distributional shift between training data and test data when solving the property prediction task.  The idea is to learn a representation that is invariant with respect to the environment as the environment change results in the distributional shift. The main assumption is that, regarding the property that we want to predict, there are specific structures that determine the property and other substructures that are not relevant (so no matter how that structure changes the property does not change ) and to avoid the distributional shift one needs to learn the representation that is invariant with respect to the change of the environment.

**Questions:**

1. The definition of distribution shift here is defined as having a different scaffold of the molecules from training to test set. In this case, is each molecule regarded as coming from a different environment as they all have different structures, it is not clear how we measure  there is a distribution shift (how different the structure of the molecule should be to be considered as there is a distributional shift? is the size of the molecules? the number of the atom, the type of the atom, to which degree)

2. The main idea is that there is spurious structure and causal structure wrt the property.  First of all, this assumption holds for most of the properties (at least the ones that they use here). Can we really identify substructures that determine the property and are totally disentangled from the rest of the structure (spurious structure)? What if the property is related to the global structure of the molecule?

If it is the case that the assumption holds,  if we have enough data and enough representation power, isn't the predictive model should learn to predict the property from the casual structure and ignoring the rest?  which means the representation will be naturally pushed to learn such invariant representation.  Unless we are really in a data scarcity case,  and if we are in the data scarcity case then learning to infer the environmental factor (unsupervised) would be also hard.

3. In equation 7, the first term and last term exactly the same except one is the sum over G^e and the other one is over G , G^e represents the graph G and y sampled from environment e, so if you take the expectation at the last term with respect to e (E_e) as you have there isn't this two-term exact the same?

4. Intuitively, it seems like what the model does is given the graph and property, learn to infer the environment, and a predictive model that learns to predict the property y for the graph given environment e (objective 6). Then use this inference and predictive model to learn another predictive model that is not restricted to the environment. But my question is if the model learned from objective 6 can infer the environment and predict the property given that environment is the problem solved? so even if we have a distributional shift we can use it to predict the property for the graph from the new environment?

**Limitations:**

The assumptions and proposed model are not really convincing.
The experiment section is a bit confusing and does not provide very structured information. For example, in table 2, what are the properties that are being predicted, and what elements are considered as the environment (still scaffold? ). If scaffold, size, and assay are considered as the environment then what is the property where it is being predicted? Also if the environment label is accessible, does the model still do inference or use it directly?

**Strengths And Weaknesses:**

strength: the problem they try to address is interesting and valuable to the community
weakness: The assumptions and proposed model are not really convincing. The experiments section is not clearly described. Hard to follow for readers who are not familiar with the dataset, so at least in appendix explains for each data set what is the property they are predicting, and what is the environment they considering here, and how the dataset is split to train test sets to model the distributional shift.

---

> ### Author Response · Authors · 2022-08-02
> **Response to Reviewer MgvC (Part 1 of 4)**
>
> Thank you for the time and valuable feedback. In the response below, we provide answers to your questions to resolve some potential misunderstandings and address the lingering points of concerns, which we hope could be helpful for the re-evaluation of our work.
>
> **Q1: How do we measure there is a distribution among molecules and how are the datasets split?**
>
> The concept of distribution in molecules datasets has reached some concensus in recent literature [1,2,3]. Specifically it is usually measured or determined by certain criteria e.g. a scaffold pattern corresponds to a certain environment whose underlying data distribution can differ from another environment with its own distribution. To be more concrete, we provide some example protocols in peer works as follows:
>
> 1. WILDS [1] provides a curated benchmark of 10 datasets reflecting a diverse range of distribution shifts, with a protocol saying: "each environment corresponds to a distribution $P_{e}$ over data points which are similar in some way, e.g. molecules with the same scaffold". In other words, for example, molecules with different scaffolds can be regarded as being sampled from different distributions.
> 2. OGB [2], a widely-used benchmark in molecule representation learning, also assumes molecules with different scaffolds are from different distributions. It should be mentioned that the official default train/val/test data split in OGB is based on scaffold splitting, which can provide a more realistic estimate of model performance in prospective experimental settings. Thus, for the four datasets BACE, BBBP, SIDER and HIV from OGB, we directly use the default data split in our experiments.
> 3. DrugOOD [3], which is a newly realeased benchmark for out-of-distribution molecule representation learning, provides two extra splitting strategies, assay and size. The original paper clearly states that molecules in the same assay or with the same number of atoms can been treated as being from the same environments, i.e., the same distribution (see Sec. 3.4.1 of DrugOOD paper). For the other six datasets we used from DrugOOD , we also adopt the official default data splits for all.
>
> The setting and used datasets (especially the four datasets from OGB) of our paper just follow the above works, and thus, to save space, we omitted some detailed descriptions for used datasets and the background information for the distribution/environment in our original version. Now we provide detailed information below and supplement them in Appendix E in the uploaded revision.
>  - **BBBP** is a dataset of Brain-Blood Barrier Penetration. Each molecule has a label indicating whether it can penetrate through brain cell membrane to enter central nervous system.
>  - **BACE** is a dataset of binding affinity against human beta-secretas 1. Each molecule has a label indicating whether it binds to human beta-secretase 1.
>  - **SIDER** is a dataset of marked drugs and adverse drug reactions (ADRs). Molecules are grouped into 27 system organ classes.
>  - **HIV** is a dataset of HIV antiviral activity. Each molecule has an active or inactive label.
>  - **IC50/EC50-scaffold/assay/size** are datasets generated by the automated dataset curator provided by DrugOOD from the large-scale bioassay deposition website ChEMBL [4]. The suffix specifies the splitting scheme. These six datasets target on ligand-based affinity prediction (LBAP). Each molecule has an active or inactive label.

---

> > ### Author Response · Authors · 2022-08-02
> > **Response to Reviewer MgvC (Part 2 of 4)**
> >
> > **Q2-1: "The main idea is that there is spurious structure and casual structure wrt the property. First of all, is this assumption holds for most of the properties (at least the ones that they use here)."**
> >
> > In general, as illustrated in our Sec 1, the hypothesis (i.e., invariance assumption) in our paper is rooted on a widely-observed phenomenon that there exist a few invariant substructures w.r.t. certain property, which is well-recognized by a surge of molecular literature across bioinformatics, pharmacy and data mining [5,6,7,8]. And, in datasets we used in the paper, such a phenomenon indeed exists. Taking **HIV** dataset as an example, *salicylhydrazide* substructure displays potent HIV-1 integrase (IN) inhibitory activity, which has been identified by previous studies [9,10]. Additionally, for **BBBP** dataset, as pointed out by recent studies [11,12], some substructures are closely related to brain-blood barrier penetration.
> >
> > On top of this, we formulate our invariance assumption in the context of molecule representation learning (i.e., existence of spurious and causal substructures wrt certain properties) as a cornerstone for theory and problem solving for OOD generalization purpose. Despite the motivation from such an invariance principle and the above evidence for the robust correlation accross environments, in fact our designed model is technically free from specific domain knowledge about substructures-property relations, like that substructure *hydroxy* displays good water solubility across all environments (which is purely used as a motivating example). To say the least, our model will not crash when this assumption does not hold, but instead it can benefit from such relation and can work smoothly as a general framework for molecule representation which can learn stable relations between some substructures and target labels in a fully data-driven manner.
> >
> > **Q2-2:"Can we really identify substructures that determine the property and is totally disentangled from the rest of the structure (spurious structure). What if the property is related to the global structure of the molecule?"**
> >
> > Thanks for your questions which are worth discussion and we will put it in our final version. It is really difficult (from the molecule science perspective) to ensure that there exist totally disentangled substructures that determine the property (on the used datasets), let alone perfectly discovering them, though this paper explores such an invariance learning direction and empirically find its effectiveness for OOD generalization.
> >
> > Technically speaking, we have proved in our theory that optimizing the new objective can guide the model to capture stable relations between environment-invariant substructures and the labels across different environments, thus ensuring a valid solution for OOD problem in principle. This result can be further justified by its consistency with other related works [13,14,15,16,17] in broad areas.
> >
> > In practice, our goal is to make the molecule encoder (which can be seen as a black-box function) to capture stable relations between environment-invariant substructures and the labels, i.e., we expect the encoder to extract causal features from input molecules to obtain the representations. The model is not designed to totally disentangle environment-invariant substructures from spurious ones. Instead, it's more like a kind of 'soft' identification for causal substructures. Since noises or biases might exist in the dataset and it's unpractical for the model to see all environments during training, it would be hard for the model to totally disentangle invariant features from spurious features in reality (though we can consider it as an ideal state to pursue for model design).
> >
> > When the property is related to the global structure of the molecule, our model design is expected to automatically discover the global structure (a special case where all substructures stably relate with the labels across environments).

---

> > > ### Author Response · Authors · 2022-08-02
> > > **Response to Reviewer MgvC (Part 3 of 4)**
> > >
> > > **Q2-3: "If it is the case that the assumption holds, if we have enough data and enough representation power, isn't the predictive model should learn to predict the property from the casual structure and ignoring the rest? which means the representation will be naturally pushed to learn such invariant representation. Unless we are really in a data scarcity case, and if we are in the data scarcity case then learning to infer the environmental factor (unsupervised) would be also hard."**
> > >
> > > For OOD generalization, the model performance is more related to the richness of environments it has seen in training instead of the quantity of data samples [13,14,15,16,17]. If the training data only contains a few environments, even though the training data is sufficient, the model is quite likely to fail to filter out irrelevant or spurious features, thus not robust to those test data from unseen environments. In contrast, if traing data involves more diverse environments, even if the number of all training data is relatively small, the model could better learn a stable relation bewteen invariant part bewteen label across environments.
> > >
> > > **Q3: Could the second term of Eqn. 7 be further simplified?**
> > >
> > > Yes. Mathematically, the second term in Eqn. 7 in our paper and $\beta\frac{1}{|\mathcal{G}|}\sum_{(G,y)\in\mathcal{G}}-\log q_\theta(y|G)$ are both equivalent to $\beta\mathbb{E}_{(G,y)}[-\log q_\theta(y|G)]$.
> > >
> > > For practical implementation, the two formulas are slightly different. The expectation $\mathbb{E}_{(G,y)}[-\log q_\theta(y|G)]$ is hard to calculate directly, thus Monte Carlo estimation is applied to approximate this value. Our implementation first uses the samples under each specific environemt for approximating the environment-specific risk and then calculate the average across different enviroments. The second term in Eqn. 7 is exactly what we have done in our implementation. Therefore, we kept this form in the paper instead of using the simplified one to stay consistent with our implementation.
> > >
> > > It should be mentioned that there is a absolute value symbol $\vert\cdot\vert$ in the first term of Eqn. 7. Hence, even if the second term of Eqn. 7 is simplified, the two terms are still completely different.
> > >
> > > **Q4: "Intuitively, it seems like what the model does is given the graph and property, learn to infer the environment, and a predictive model that learns to predict the property y for the graph given environment e (objective 6). Then use this inference and predictive model to learn another predictive model that is not restricted to the environment. But my question is if the model learned from objective 6 can infer the environment and predict the property given that environment is the problem solved? so even if we have a distributional shift we can use it to predict the property for the graph from he new environment?"**
> > >
> > > Only using environment inference model learned from the objective in Eqn. 6 is insufficient for solving the challenging OOD problem. The reasons are as follows. First, during training stage, the environment inference model is to partition the training data into $k$ environments. But in out-of-distribution problem, the environments of testing data are often unseen during training. Therefore, the well-trained environment inference model could not properly map the testing instance to those $k$ training environments. Second, the environment classifier requires the label $y$ as its input to preidict the corresponding environment. But for the testing data, label $y$ is not available and exactly what we need to predict.
> > >
> > >
> > > We hope this response could help to address your concerns. As we believe, our work is one of the early efforts to study an important problem, out-of-distribution molecule representation learning, with novel methodology and promising results. We sincerely hope that you could reconsider your assessment.

---

> > > > ### Author Response · Authors · 2022-08-02
> > > > **Response to Reviewer MgvC (Part 4 of 4)**
> > > >
> > > > **Reference (only [4,9,10,11,12] are new and will be added in the referene of the paper):**
> > > >
> > > > [1] [WILDS: A Benchmark of in-the-Wild Distribution Shifts.](http://proceedings.mlr.press/v139/koh21a/koh21a.pdf)
> > > >
> > > > [2] [Open Graph Benchmark: Datasets for Machine Learning on Graphs.](https://arxiv.org/pdf/2005.00687.pdf)
> > > >
> > > > [3] [DrugOOD: Out-of-Distribution (OOD) Dataset Curator and Benchmark for AI-aided Drug Discovery -- A Focus on Affinity Prediction Problems with Noise Annotations.](https://arxiv.org/pdf/2201.09637.pdf)
> > > >
> > > > [4] [ChEMBL: towards direct deposition of bioassay data.](https://pdfs.semanticscholar.org/5584/f0f28d6054fd04ec9d8d066b67966825fa54.pdf)
> > > >
> > > > [5] [Chemical substructures that enrich for biological activity](https://academic.oup.com/bioinformatics/article/24/21/2518/192573)
> > > >
> > > > [6] [Privileged substructures for anti-sickling activity via cheminformatic analysis](https://pubs.rsc.org/en/content/articlehtml/2018/ra/c7ra12079f)
> > > >
> > > > [7] [DGDFS: Dependence Guided Discriminative Feature Selection for Predicting Adverse Drug-Drug Interaction](https://ieeexplore.ieee.org/abstract/document/9023472)
> > > >
> > > > [8] [A substructure-based screening approach to uncover N-nitrosamines in drug substances](https://pubmed.ncbi.nlm.nih.gov/35647726/)
> > > >
> > > > [9] [Discovery of novel non-cytotoxic salicylhydrazide containing HIV-1 integrase inhibitors.](https://www.sciencedirect.com/science/article/abs/pii/S0960894X07011559)
> > > >
> > > > [10] [Salicylhydrazine-Containing Inhibitors of HIV-1 Integrase:  Implication for a Selective Chelation in the Integrase Active Site.](https://pubs.acs.org/doi/abs/10.1021/jm9801760)
> > > >
> > > > [11] [Estimation of ADME Properties with Substructure Pattern Recognition.](https://pubs.acs.org/doi/abs/10.1021/ci100104j)
> > > >
> > > > [12] [A classification model for blood brain barrier penetration.](https://www.sciencedirect.com/science/article/pii/S1093326319303547)
> > > >
> > > > [13] [Invariant Risk Minimization.](https://arxiv.org/pdf/1907.02893.pdf)
> > > >
> > > > [14] [Invariance, causality and robustness.](https://projecteuclid.org/journals/statistical-science/volume-35/issue-3/Invariance-Causality-and-Robustness/10.1214/19-STS721.full)
> > > >
> > > > [15] [Invariant models for causal transfer learning.](https://www.jmlr.org/papers/volume19/16-432/16-432.pdf)
> > > >
> > > > [16] [Out-of-distribution generalization via risk extrapolation (rex).](http://proceedings.mlr.press/v139/krueger21a.html)
> > > >
> > > > [17] [Environment inference for invariant learning.](https://proceedings.mlr.press/v139/creager21a.html)

---

> > > > > ### Author Response · Authors · 2022-08-07
> > > > > **Look Forward to Feedbacks**
> > > > >
> > > > > Dear Reviewer MgvC,
> > > > >
> > > > > Thanks again for your time and thorough review.
> > > > >
> > > > > In our early response, we have included detailed answers to your questions in the initial review. It would be grateful if you can confirm whether our response has addressed your concerns.
> > > > >
> > > > > If you have any further questions, please let us know, so that we can provide follow-up response timely.
> > > > >
> > > > > Sincerely, Authors

---

> > > > > ### Author Response · Authors · 2022-08-09
> > > > > **Inquiry for post-rebuttal comments**
> > > > >
> > > > > Dear Reviewer MgvC,
> > > > >
> > > > > Thanks again for your time and valuable comments. Since the discussion deadline is approaching, we would be glad to hear from you whether our response has addressed your concerns.
> > > > >
> > > > > Sincerely, Authors

---

### Official Review · Reviewer_pgdS · 2022-07-10

**Rating:** 6
**Confidence:** 4
**Soundness:** 3 good
**Presentation:** 2 fair
**Contribution:** 3 good

**Summary:**

## Summary

This paper introduces techniques to enhance the robustness of molecule representation learning against distribution shifts. The authors made the observation that bio-chemical properties of molecules are usually invariantly associated with certain substructures across different environments such as scaffolds, sizes, etc. There are three important pieces in their modeling process. One is architecture-wise they use graph-level embeddings to attend a list of substructures. Second they device a new learning objective from mutual information and invariant learning to help select causal substructures. Third they use a latent representation for environment to mitigate the current issue of dealing molecular environments (e.g., often man-made, not always available, too many if using scaffolds). Overall, it's a solid paper.

**Questions:**

## Questions

- what's the exact definition of $\mathcal{G}^e$, can you add text and math formula to explain?
- it seems that the second term of eqn 7 could be simplified to $\beta \frac{1}{|\mathcal{G}|} \sum_{(G, y) \in \mathcal{G}} -log q_{\theta}(y | G)$, can you confirm?
- BBBP benchmark ROC seems much lower than typical methods would produce, is it some mistake? reference: https://arxiv.org/abs/2111.12951 appendix G table s1.
- Often times bio-chemical properties are affected by interactions between substructures, would be interesting to see if adding self-attention in the molecule encoder would help

**Strengths And Weaknesses:**

## Strengths
- the paper addresses an important topic in molecule representation learning (distribution shift).
- it introduces interesting innovative techniques to help mitigate issues of current methods

## Weaknesses
- some key notations are not clearly defined - care is needed for the writing

---

> ### Author Response · Authors · 2022-08-02
> **Response to Reviewer pgdS (Part 1 of 2)**
>
> Thank you for the valuable comments and suggestions. We are encouraged that you appreciated our technical contributions including the problem significance, novelty, soundness and solid experiments. Below we respond to your specific comments.
>
> **Q1: Missing exact definition of $\mathcal{G}^{e}$.**
>
> Thanks for pointing this out. $\mathcal{G}^{e}$ denotes the set of graph instances (each consisting of a molecule $G$ and corresponding label $y$) under environment $e$, i.e., $\mathcal{G}^{e}= \lbrace(G,y)|(G,y)\sim p(\mathbf{G},\mathbf{y}\vert\mathbf{e}=e)\rbrace$. We have supplemented this definition in our revised paper.
>
> **Q2: Could the second term of Eqn. 7 be further simplified?**
>
> Idealy, the second term in Eqn. 7 in our paper and $\beta\frac{1}{|\mathcal{G}|}\sum_{(G,y)\in\mathcal{G}}-\log q_\theta(y|G)$ are both mathematically equivalent to the simplified form $\beta\mathbb{E}_{(G,y)}[-\log q_\theta(y|G)]$.
>
> Yet for implementation, the two formulas are slightly different. The expectation $\mathbb{E}_{(G,y)}[-\log q_\theta(y|G)]$ is hard for direct computation, thus we use Monte Carlo estimation for approximation. Our implementation first uses the samples under each specific environemt for approximating the environment-specific risk and then calculate the average across different enviroments. The second term in Eqn. 7 is exactly what we have done in our implementation. Hence, we kept this form in the paper instead of using the simplified one to stay consistent with our implementation.
>
> **Q3: Lower baseline performance on BBBP benchmark compared to the mentioned paper.**
>
> Our experiments of baselines (GCN, GIN and GraphSAGE) are conducted under the official default train-valid-test data split given by OGB benchmark, using the implementation provided by OGB (see Appendix C). According to the OGB original paper [1], the dataset is split by scaffold, which already fits the OOD setting. Our final results of baselines on BBBP are consistent with those demonstrated in OGB original paper (see Table 24 in Appendix A of [1]).
>
> For the compared baselines, to our best knowledge, please note that only the mentioned paper shows a higher performance than ours. For example, the baselines in the NeurIPS'21 paper [2] also shows close performance to ours in terms of ROC-AUC. Thus, we believe we have tried our best in making a fair comparison.
>
> We noticed that the detailed experimental settings on BBBP seems to be unclearly presented in the mentioned paper, and even with authors' codes publicly released, the original detailed splitting information is missing. We suspect that they adopted a different train/val/test data split rather than the widely-used default split of OGB.

---

> > ### Author Response · Authors · 2022-08-02
> > **Response to Reviewer pgdS (Part 2 of 2)**
> >
> > **Q4: Incorporating the idea that bio-chemical properties may be affected by interactions between substructures into the design of the Molecule Encoder.**
> >
> > To verify your hypothsis, we supplement new results of our tentative exploration in the table below. To encode interactions between substructures into the final learned molecular representation, we utilize the permutation equivariant Set Attention Block (SAB) proposed in Set Transformer [3]. SAB takes a representation set of any size as input and outputs a representation set of equal size. SAB is able to encode pairwise and higher-order interactions between elements in input sets into outputs. We add such a SAB after the Substructure Encoder. For each molecule, we feed the representions of its substructures to SAB to obtain new substruture representations. In this way, the final molecule representation could model interactions between substructures. Due to limited time, we only conduct experiments on EC50-assay/scaffold/size to examine the performance of adding such a SAB. As demonstrated in the table, we can see that adding such a SAB further improves our model on EC50-scaffold. This design is a naive attempt but brings us some valuable insights. We can put the current results in appendix and leave further exploration for future directions.
> >
> > |               |       **EC50-assay**       |     **EC50-scaffold**      |       **EC50-size**        |
> > |:------------- |:--------------------------:|:--------------------------:|:--------------------------:|
> > | **ERM**       |       $69.35\pm7.38$       |       $63.92\pm2.09$       |       $60.94\pm1.95$       |
> > | **IRM**       |       $69.94\pm1.03$       |       $63.74\pm2.15$       |       $58.30\pm1.51$       |
> > | **DeepCoral** |       $69.42\pm3.35$       |       $63.66\pm1.87$       |       $56.13\pm1.77$       |
> > | **DANN**      |       $66.97\pm7.19$       |       $64.33\pm1.82$       |       $61.11\pm0.64$       |
> > | **MixUp**     |       $70.62\pm2.12$       |       $64.53\pm1.66$       |       $62.67\pm1.41$       |
> > | **GroupDro**  |       $70.52\pm3.38$       |       $64.13\pm1.81$       |       $59.06\pm1.50$       |
> > | **Ours**      |  $\mathbf{73.25\pm1.24}$   | $\underline{66.69\pm0.34}$ |  $\mathbf{65.09\pm0.90}$   |
> > | **Ours+SAB**  | $\underline{73.15\pm2.69}$ |  $\mathbf{67.26\pm1.54}$   | $\underline{64.83\pm1.07}$ |
> >
> > **Reference (only [2,3] are new and [3] will be added in the reference of the main paper):**
> >
> > [1] [Open Graph Benchmark: Datasets for Machine Learning on Graphs.](https://arxiv.org/pdf/2005.00687.pdf)
> >
> > [2] [Graph Adversarial Self-Supervised Learning.](https://proceedings.neurips.cc/paper/2021/file/7d3010c11d08cf990b7614d2c2ca9098-Paper.pdf)
> >
> > [3] [Set Transformer: A Framework for Attention-based Permutation-Invariant Neural Networks.](http://proceedings.mlr.press/v97/lee19d/lee19d.pdf)

---

### Author Response · Authors · 2022-08-06
**General Response by Authors**

We thank the reviewers for their time and valuable comments. Overall, the reviewers found our work well-motivated (WE4q) and novel (pgdS, WE4q), and appreciated the clear and well-organized presentation (WE4q), technically soundness (pgdS, WE4q), extensive and thorough experiments (WE4q), as well as good connection with NeurIPS community (MgvC, WE4q). To facilitate the reviewing process towards a comprehensive evaluation of our work, we first restate our contributions below:

- **Methodology** We leverage the invariance principle as an effective prior and devise a new learning objective to learn robust molecular representations for out-of-distribution generalization purpose. To our knowledge, we are the pioneering work in this direction. Also, our model is free from manual specifications of environments and can incorporate off-the-shelf molecular encoders to improve their robustness against distribution shifts.
- **Theoretical analysis** We also provide theorectical analysis to back up our proposed method. Theoretical justifications reveal that optimizing the proposed objective forces the learned molecular representation to satisfy the invariance principles, thus guaranteeing a valid solution for OOD problem.
- **Empirical performace** We conduct extensive and comprehensive on ten publicly available datasets. Results demonstrate that our proposed model shows a superior generalization ability than state-of-the-art models. In particular, our method achieves up to 5.9% and 3.9% improvement over the strongest baselines on OGB and DrugOOD benchmarks in terms of ROC-AUC, respectively.

We will sincerely appreciate it if you could post some comments so that we can improve this paper accordingly.

---

### Comment · Area_Chair_mm4L · 2022-08-07
**Discussion with Authors**

Dear Reviewers! Thank you so much for your time on this paper so far.

The authors have written a detailed response to your concerns. How does this change your review?

Please engage with the authors in the way that you would like reviewers to engage your submitted papers: critically and open to changing your mind.

Looking forward to the discussion!

---

### Author Response · Authors · 2022-08-10
**Look forward to feedbacks**

Dear Reviewers,

Thanks again for your valuable comments and nice suggestions. We are still sincerely looking forward to your feedbacks. It is really a good chance for us to engage in the discussion to help us improve this paper.

Sincerely, Authors

---

### Meta-Review · Area_Chair_mm4L · 2022-08-26

**Recommendation:** Accept
**Confidence:** Less certain

**Metareview:**

All reviewers agreed that this paper should be accepted because of the strong author response during the rebuttal phase. Specifically the reviewers appreciated the motivation of the paper, its clarity, and added explanation and experiments included during the rebuttal. Authors: please carefully revise the manuscript based on the suggestions by the reviewers: they made many careful suggestions to improve the work and stressed that the paper should only be accepted once these changes are implemented. To these suggestions I urge the authors to add another: I strongly suggest removing section 3.2. This data generating process is not validated and is not at all necessary for your approach. The SCM is never referred to again outside of this section. All that is necessary is that one can view molecules as coming from different environments or contexts and predicting this context is useful to improve generalization. Finally, with the added space I suggest expanding figure 1 to add more examples of “environments” and make this clearer in the figure: right now you only mention briefly in the caption that different scaffolds can be thought of as different environments. If you could include more / better examples that align with your experiments this will make the motivation clearer. Once these changes are made the paper will be a nice addition to the conference!

**Award:**

No

---

### Decision · Program_Chairs · 2022-09-14

Accept